# MULTI-CRITIC ACTOR LEARNING:
# TEACHING RL POLICIES TO ACT WITH STYLE

**Siddharth Mysore**
Boston University & Electronic Arts

**George Cheng & Yunqi Zhao**
Electronic Arts

**Kate Saenko**
Boston University & MIT-IBM Watson AI Lab

**Meng Wu**
Electronic Arts

## ABSTRACT

Using a single value function (critic) shared over multiple tasks in Actor-Critic multi-task reinforcement learning (MTRL) can result in negative interference between tasks, which can compromise learning performance. Multi-Critic Actor Learning (MultiCriticAL) proposes instead maintaining separate critics for each task being trained while training a single multi-task actor. Explicitly distinguishing between tasks also eliminates the need for critics to learn to do so and mitigates interference between task-value estimates. MultiCriticAL is tested in the context of multi-style learning, a special case of MTRL where agents are trained to behave with different distinct behavior styles, and yields up to 56% performance gains over the single-critic baselines and even successfully learns behavior styles in cases where single-critic approaches may simply fail to learn. In a simulated real-world use case, MultiCriticAL enables learning policies that smoothly transition between multiple fighting styles on an experimental build of EA's UFC game.

## 1 INTRODUCTION

Reinforcement Learning (RL) offers an interesting means by which to develop interesting behaviors in a variety of controls settings. The work presented in this paper is primarily contextualized by our efforts to develop RL techniques for teaching RL-policies to behave with multiple distinct and specific styles for game-play. RL can be useful in developing control policies for game-play and has been demonstrated as being capable of learning human-like or even superhuman play, most notably beating top-ranked human players in games such as Dota 2 (Berner et al., 2019), StarCraft (Vinyals et al., 2019), Chess (Silver et al., 2017), and Go (Silver et al., 2016). RL-based control in games offers a wide array of potential applications, including testing (Zhao et al., 2020; Ariyurek et al., 2021; Gordillo et al., 2021), game design (Gisslén et al., 2021), providing competition for human players (Berner et al., 2019), or simply as a means to develop more interesting game-play AI behaviors (Zhao et al., 2020; Alonso et al., 2020). RL algorithms can be notorious however for training aesthetically unappealing policies, but prior work has demonstrated that, with careful shaping of the reward functions, it is possible to derive highly specific desirable behavior (Peng et al., 2018; Mysore et al., 2021b). Machine learning (ML) solutions to game-play AI can however represent a significant increase in compute requirements over their heuristic-based counterparts, especially when using deep learning techniques. If a learned control policy could represent multiple desirable styles and offered a controllable way to transition between the learned styles, this could reduce the total compute burden of the ML-based controller, making it more practically viable. Ideally, developing an ML model would constitute learning a single, low-complexity policy that would model and transition smoothly between multiple behavior styles. Reduced model complexity and deploying just a single policy network enables greater compute efficiency through hardware parallelism and reduced memory costs, which is beneficial in resource-constrained applications, such as videogames.

We explore the problem of *mutli-style* RL, a special case of the more commonly explored problem of multi-task RL (MTRL). Whereas MTRL techniques typically seek to solve a wide array of RL problems under a single learning campaign, multi-style RL adds the extra constraint that the 'tasks' in question all have identical system dynamics but would be characterized by different re-

Figure 1: MultiCriticAL breaks from the common practice of using a single unified critic in MTRL and instead uses separate critics for each task learned. The proposed method is used to successfully train multiple distinct behavior styles in various games including Pong and UFC.

ward signals corresponding to the different desired styles of behavior. We focus specifically on applications of Actor-Critic RL techniques, which, in addition to being compatible with continuous action-space control, allow for reduced runtime computational cost as only the actor functions are required for inference, while the critics can often be disregarded after training. Mysore et al. (2021a) and Andrychowicz et al. (2021) have also demonstrated that it is possible to train compact and performant policy networks through careful consideration of the actor and critic network architectures.

A common baseline approach to MTRL is to employ one-hot task encoding to delineate between tasks. However, in a number of multi-style cases considered this was not sufficient to enable successful multi-style policy-learning. We suspected that the similarity of game-play states visited between each trained style interfered with each style's value optimization. In surveying existing MTRL literature for advancements in the field of multi-task Actor-Critic RL methods, we noticed a surprising hole. There are studies that employ single actors and single critics (Finn et al., 2017; Yang et al., 2020; Sodhani et al., 2021; Nichol et al., 2018), multiple actors and multiple critics (Andreas et al., 2017; Teh et al., 2017; Rusu et al., 2015; 2016; Huang et al., 2017), or multiple actors and a single critic (Yang et al., 2017; Dewangan et al., 2018). Curiously, seemingly no prior work explores using a single policy with distinct, per-task value estimation, i.e. a single actor with multiple critics.

This paper proposes Multi-Critic Actor Learning (MultiCriticAL), a single-actor, multi-critic framework for multi-task Actor-Critic optimization, and evaluates it on a variety of multi-style learning problems. The core idea of MultiCriticAL is to maintain separate per-style (or task) critic functions for each style being learned. By separating the critic MultiCriticAL would be able to avoid negative interference in the learned values for different styles. Our results show that MultiCriticAL consistently trains more performant policies, compared to its single-critic counterparts, succeeds in cases where the single-critic methods fail and achieves between 20-45% improvement in more traditional multi-task settings involving multi-level game-play. Additionally, we demonstrate the utility of MultiCriticAL in a use-case more reflective of real-world application, in an experimental build for EA's UFC Knockout Mode, where we train policies to act with multiple specific and distinct behavior styles. This work is also amongst the first to study the efficacy of MTRL algorithms across a broad range of multi-style problems, an aspect of MTRL receiving limited prior attention.

## 2 PRELIMINARIES

The core reinforcement learning (RL) problem is framed as a finite-horizon Markov decision process, $\langle S, A, P, r, \gamma \rangle$, with state transition dynamics $P : S, A \rightarrow S$, for states $s \in S$ and actions $a \in A$, and reward function $r : S, A \rightarrow \mathbb{R}$. The RL objective is to train a policy, $\pi : S \rightarrow A$ such that the expected sum of discounted rewards, $\mathbf{E}_{\tau_\pi} \left[ \sum_i \gamma^i r(s_i, a_i) \right]$, with discount factor $\gamma$, is maximized for all trajectories, $\tau_\pi$, sampled on policy $\pi$. RL typically assumes that both the transition dynamics and reward dynamics are encompassed by a black-box environment. RL agents interact with the environment to observe the state response and generated rewards, but where specific reward functions and transition dynamics are not assumed known. Solving the RL optimization problem requires estimating expected rewards and using these approximations to optimize the action policies.

Among the different techniques for tackling RL optimization, we mainly consider Actor-Critic methods, whose key characteristic is their separation of the 'actors', representing the RL policy, and the 'critic', representing the value-function estimator that learns to estimate future rewards. This sep-

aration allows Actor-Critic RL to take advantage of the improved sample-efficiency of value-based techniques, such as Q-learning (Watkins & Dayan, 1992) while taking advantage of policy-based techniques, such as policy gradients (Sutton et al., 2000), for improved robustness to stochasticity and to enable learning in continuous action domains. Examples of prominent contemporary algorithms in this class of RL methods include TRPO (Schulman et al., 2015), PPO (Schulman et al., 2017), DDPG (Lillicrap et al., 2016), TD3 (Fujimoto et al., 2018), and SAC (Haarnoja et al., 2018).

In algorithms based on Q-learning, such as DDPG, TD3, and SAC, the policy optimization criteria $J_{\pi_\theta}$ for a policy, $\pi$, parameterized by $\theta$, is proportional to the Q-value:

$$J_{\pi_\theta} \propto Q^{\pi_\theta}(s,a) = \mathbf{E}\left[R(\tau_{\pi_\theta})|s_0 = s, a_0 = a\right] = \mathbf{E}_{s'}\left[r(s,a) + \gamma Q^{\pi_\theta}(s', \pi_\theta(s'))\right] \quad (1)$$

where $s'$ is the next state reached from state $s$ in response to action $a$, and $R(\tau)$ is the discounted sum of rewards over trajectory $\tau$. For algorithms based on a more traditional policy gradient optimization, such as TRPO and PPO, policies are optimized proportional to the advantage function, $A^{\pi_\theta}$ and a measure of policy divergence:

$$J_{\pi_\theta} \propto \log(\pi_\theta(a|s))A^{\pi_\theta}(s,a) = \log(\pi_\theta(a|s))\left(Q^{\pi_\theta}(s,a) - V^{\pi_\theta}(s)\right) \quad (2)$$

where the value-function $V^\pi(s) = \mathbf{E}_{\tau_\pi}\left[R(\tau_\pi)|s_0 = s\right] = \mathbf{E}_{s'}\left[r(s, \pi(s)) + \gamma V^\pi(s')\right]$. For critic networks parameterized by $\phi$, either a state-value estimator, $V_\phi$, or Q-value (state-action-value) estimator, $Q_\phi$ are optimized with optimization criteria $J_{V_\phi^\pi}$ or $J_{Q_\phi^\pi}$ respectively:

$$J_{V_\phi^\pi} \propto ||V_\phi^\pi(s) - (r(s, \pi(s)) + V_\phi^\pi(s' \sim P(s, \pi(s))))|| \quad (3)$$

$$J_{Q_\phi^\pi} \propto ||Q_\phi^\pi(s,a) - (r(s,a) + Q_\phi^\pi(s' \sim P(s,a), \pi(s')))|| \quad (4)$$

While specific details around how actors and critics are optimized vary between algorithms, the above represents the fundamentals of optimization criteria upon which we shall build our discussion.

## 3 RELATED WORK

We broadly categorize multi-task RL (MTRL) techniques as either *single-policy*, if the same policy function activations are used to determine actions over all tasks, or *multi-policy*, if more separate activations (in the case of multi-headed policies) or policy functions are used. Multi-policy techniques range from training multiple policies and selecting or mixing between them (Andreas et al., 2017; Peng et al., 2018; Dewangan et al., 2018; Yang et al., 2017; Huang et al., 2017), to simultaneously or iteratively training additional policies while sharing representation between policies (Rusu et al., 2015; Teh et al., 2017; Rusu et al., 2016). Techniques requiring multiple policy networks are limited by their storage and computational costs during inference, and multi-headed policies can limit our ability to smoothly transition between behaviors, which makes them less attractive in the context of RL for game-play. Therefore, we focus instead on single-policy learning.

Single-policy MTRL techniques train a single policy function to solve multiple tasks, without relying on information from previously trained policies. Model-free meta-learning techniques have been proposed to improve multi-task generalization, in the absence of task specificity (Finn et al., 2017; Nichol et al., 2018), but where explicit task distinction can be exploited, a common approach has been to exploit task encoding to distinguish between tasks. A simple, yet effective approach to this problem has been to extend state representations with one-hot task encoding (Peng et al., 2018; Yu et al., 2020; Yang et al., 2020), thus allowing policies to distinguish between individual tasks and decide actions in a task-appropriate way. Recent work has also sought to enhance the performance and generalizability of such single-policy techniques through automated task encoding (Sodhani et al., 2021) or through soft-modularization (Yang et al., 2020), which encourages the development of distinct information pathways per task within actor networks.

Our work considers the special MTRL case of *multi-style* learning. Multi-style learning requires agents to learn different behaviors for highly similar states while providing potentially highly dissimilar rewards for each style. This variability in the reward signals can have an adverse impact on the quality of learned value-functions when using a single value estimator if the estimator cannot sufficiently distinguish between styles. We hypothesize that *negative interference* more strongly impacts value-function learning in multi-style tasks as task similarity may make specific style-values harder to distinguish. Multi-style learning is however an aspect of MTRL that has not received much

prior consideration. Peng et al. (2018) explore the most similar setting to our work – where agents were required to learn different movement styles in animation tasks. Their work tested single-policy MTRL and found that policies were only able to learn a limited number of styles, requiring the authors to resort to a multi-policy approach.

Most works considering the Actor-Critic MTRL problem focus predominantly on how the actors, i.e the policy-functions, are optimized, with less attention given to critics. The problem of negative interference between tasks in the broader context of MTRL is one that has been highlighted and studied before (Yang et al., 2020; Parisotto et al., 2015; Teh et al., 2017; Standley et al., 2020) however, but has similarly been approached as a policy optimization problem. A number of multi-policy techniques were noted to maintain separate valuations per policy (Teh et al., 2017; Andreas et al., 2017; Peng et al., 2018), though some work has also suggested that sharing critics can allow optimization to better capture useful shared knowledge to improve policy optimization (Dewangan et al., 2018; Yang et al., 2017). A similar idea dominates single-policy techniques, where all the studied methods maintain a single value-function estimator across all tasks (Nichol et al., 2018; Yu et al., 2020; Peng et al., 2018; Yang et al., 2020; Sodhani et al., 2021). While task-encoding may allow separate task-values to be learned implicitly by the critics, this was never found to be explicitly designed. Our work alternatively frames the multi-style learning framework as one that explicitly learns distinct per-style values to propose a value-learning optimization scheme that is better able to accommodate disparate reward functions.

## 4 MULTI-CRITIC ACTOR LEARNING

The policy functions, i.e. the 'actor' part of Actor-Critic frameworks, represent the core decision-making processes of RL agents and the success or failure of any algorithm rests on the policies' abilities to learn useful mappings. The learned critics have limited utility after training and learned value-function estimators are also known for being poor approximations of the true value-functions they attempt to capture (Ilyas et al., 2020), so it makes sense that prior work has focused more on how policies are constructed and trained. The limitations of value learning are however exactly why the specifics of value-network constructions and optimization are important to consider. A critical element in the performance of RL algorithms can be the representational capacity of the learned value-functions (Mysore et al., 2021a; Andrychowicz et al., 2021). It, therefore, stands to reason, that better structuring how value-networks represent the learned values can be an important component of improving value-function learning and therefore policy performance.

A common formulation of single-critic MTRL optimization involves including a task-encoding, $\Omega$, to distinguish between learned tasks and to specify task rewards and updating the state representation $\bar{s} = \langle s, \Omega \rangle$. The optimization criteria in Equations 1-4 are correspondingly updated to use $\bar{s}$ in place of $s$. The resulting MTRL optimization criteria typically represent rewards and learned values by scalar values. Practically, the value-functions are learned by a single network with a single output node, regardless of the number of tasks. A single learned value-function representing multiple values may enforce continuity between the learned values, which may not necessarily reflect the true value functions and could compromise the quality of learned values and resulting policies.

**Value Optimization for Multi-Critic Actor Learning**    By separating the learned values per style, assumptions of continuity between style-values can be dismissed and style (or task) values can be learned explicitly, without relying on the critics to inherently learn the distinctions. We propose Multi-Critic Actor Learning (MultiCriticAL) with a style-conditioned value optimization criteria:

$$J_{\mathbf{V}_\phi^\pi|\Omega} \propto \left|\left|\sum_i \omega_i \left[V_{\phi_i}^\pi(s) - (r_i(s, \pi(\bar{s})) + V_{\phi_i}^\pi(s' \sim P(s, \pi(\bar{s}))))\right]\right|\right| \tag{5}$$

$$J_{\mathbf{Q}_\phi^\pi|\Omega} \propto \left|\left|\sum_i \omega_i \left[Q_{\phi_i}^\pi(s, a) - (r_i(s, a) + Q_{\phi_i}^\pi(s' \sim P(s, a), \pi(\bar{s}')))\right]\right|\right| \tag{6}$$

where, for $M$ styles, the task/style encoding, $\Omega = [\omega_0, \ldots, \omega_M]$ and $\omega_i \in [0, 1]$. The policy optimization criteria requires the value functions to be continuous with respect to the states and task encoding, and this property can be maintained by appropriately selecting the value functions when computing the policy optimization criteria. The specifics of the exact formulation of the optimization criteria would vary in accordance with the base Actor-Critic algorithm employed, with Equa-

tions 1 and 2 respectively being updated as:

$$J_{\pi_\theta} \propto \sum_i \omega_i Q_{\phi_i}^{\pi_\theta}(s, a) \tag{7}$$

$$\text{or } J_{\pi_\theta} \propto \log(\pi_\theta(a|\bar{s})) \sum_i \omega_i A_{\phi_i}^{\pi_\theta}(s, a) \tag{8}$$

Practically, multi-valued critics may be represented by separate independent critic networks – multi-network (MN) MultiCriticAL – or a single base critic network with multiple heads, where each head represents a different learned value – multi-headed (MH) MultiCriticAL. While the latter allows for some learned representations to be shared between the value functions, the former may be more beneficial when sharing is likely to lead to undesirable interference. This is visually shown in Figure 1 and extended in Figure 14 in Appendix C.

**Relationship to MTRL and MORL DQNs**  MultiCriticAL shares theoretical underpinnings with existing DQN techniques to tackle MTRL and multi-objective RL (MORL) by learning task/objective-specific values (Parisotto et al., 2015; Moffaert & Nowé, 2014; D'Eramo et al., 2020; Rusu et al., 2016). Learned DQN policies and value functions share a single combined architectural backbone, forcing value estimation machinery to be retained at inference time. By extending the theory to Actor-Critic methods, we both enable utility in continuous control and the practical benefits of disentangling the representational complexity of the actors from the often greater representational burden placed on the critics (Mysore et al., 2021a). (Additional discussion in Appendix B).

## 5 EVALUATION

We evaluate the proposed MultiCriticAL method in a series of multi-style RL problems. We first test our method on two basic multi-style learning environments designed as simple and light-weight demonstrations of the types of problems we seek to address: (i) learning to follow simple shapes, where different styles correspond to different shapes, (ii) playing a modified game of Pong where the player's paddle is allowed to rotate and agents are trained to play aggressively or defensively against a stock AI. While relatively simple, these environments help highlight the efficacy of MultiCriticAL over the more typical single-critic MTRL framework. Additionally, we test MultiCriticAL on learning to play multiple levels of the Sega genesis games Sonic the Hedgehog, and Sonic the Hedgehog 2, where the agent is trained to play the different levels of each game, following the reward structure devised for the Gym Retro contest (Nichol et al., 2018). Finally, we demonstrate a practical use-case for MultiCriticAL in an experimental build of EA's UFC Knockout Mode fighting game where RL agents are integrated into the game controls and trained with different fighting styles. Appendix A provides learning curves and additional visualizations for the results discussed in this section.

**Baselines**  We train MultiCriticAL with two popular contemporary actor-critic algorithms: PPO (Schulman et al., 2017) and SAC (Haarnoja et al., 2018), which represent Actor-Critic algorithms based on both value iteration and Q-learning respectively. These are compared against Multi-task PPO (MTPPO) and Multi-task SAC (MTSAC), which augment the state information with a one-hot task encoding and were also proposed as baselines by Yu et al. (2020). While we recognize that these are not state-of-the-art MTRL algorithms, they share a common critic optimization structure with more recent advancements in policy (actor) optimization (Sodhani et al., 2021; Yang et al., 2020), and we aim to demonstrate the fundamental utility of MultiCriticAL in the absence of additional policy optimization tricks. Training code is based on OpenAI's Spinning Up (Achiam, 2018) and is provided in the Supplementary Material for the multi-style environments. As the Sonic the Hedgehog games require paid access and the development environment for UFC is proprietary, game code is not provided, though Appendix D details how they were adapted.

**Seeding**  To account for the variability in RL, performance is averaged over 15 random seeds.

### 5.1 PATH FOLLOWING

**Environment Design and Reward Shaping**  In the path following environment, given its current position in space, the current phase of motion $\in [0, 1]$, and an indicator on which shape to reproduce, the agent is expected to follow a trajectory that generates the desired shape. Three shapes are tested:

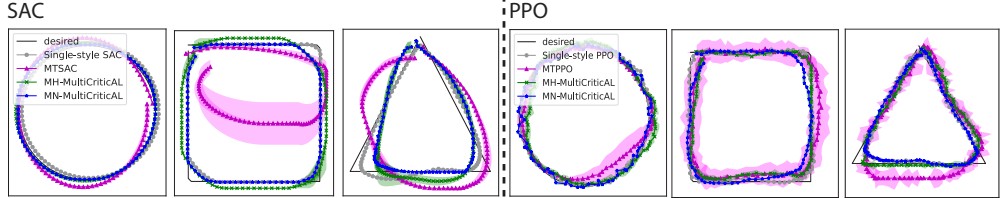

Figure 2: The path-following task requires agents to learn a Circular, Square, or Triangular path. MultiCriticAL agents can learn to follow all 3 shapes with better performance compared to the more typical single-critic multi-task value-learning frameworks of MTSAC and MTPPO.

Circle, Square and Triangle. This simple problem adds no additional complexity related to generalization or complex system dynamics and simply expects policies to learn a mapping function that encodes the desired shape. The multi-style objective is for the agents to accurately produce any of the three required shapes, given the task encoding. Agents are rewarded based on the absolute error between the agent's current and desired position in space. The rewards are cast to be positive in the range of $[0, 1]$ per step. Appendix D offers further details, with code in Supplementary Material.

**Results**   The pathing behavior learned for each algorithm over the three shapes is shown in Figure 5.1. While single-style SAC and PPO are able to learn to follow each shape, performances of MTSAC and MTPPO in the multi-style setting are compromised. This is most prominent with MTSAC. MTPPO fares better but performance in the multi-style setting is still notably worse. Adapted to both algorithms, MultiCriticAL more consistently learns to follow all three shapes. It should also be noted that larger networks were needed for the PPO algorithm to solve this task, using two hidden layers with 32 neurons each, as opposed to SAC's 8 each. This likely contributes to the improved performance of PPO in the multi-task setting as the networks being used have higher relative representational capacity, but are still outperformed by MultiCriticAL.

## 5.2   PONG WITH ROTATABLE PADDLES

**Environment Design and Reward Shaping**   We use a modified game of Pong to explore a multi-style learning problem where one style is represented by a sparse reward, and the other a dense reward, anticipating that this might be a case where single-critc value-function estimation may fail. Pong typically allows only a single degree of freedom for paddle movement and does not normally command nuanced strategies. Adding an extra degree of freedom by allowing the player's paddles to rotate introduces increased complexity to the game and facilitates the training of different behavior styles. For this game, we consider two primary styles of play: (i) aggressive – with the objective of winning the game as quickly as possible, and (ii) defensive – where the objective is to avoid losing while prolonging the game as long as possible. In a real game setting, these could be analogous to hard- and easy-mode AI respectively, where the former tries to give the opponent a challenge and the latter may help the opponent learn to play. Our implementation is based on the Pong environment provided by the Pygame Learning Environment (PLE) (Tasfi, 2016). For simplicity, the basic opponent AI cannot rotate its paddle and uses a heuristic-based Pong controller provided by PLE. The *Aggressive* style rewards winning as quickly as possible while the *Defensive* style encourages avoiding losing and ensuring that the opponent can keep receiving the ball to prolong the game (additional details in Appendix D).

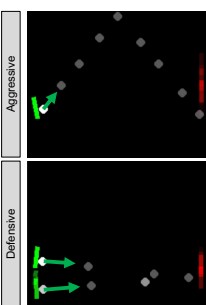

Figure 3: Behavior samples for *Aggressive* and *Defensive* play in Pong.

**Results**   We first established baseline performance by training SAC and PPO on the *Aggressive* and *Defensive* play-styles. Figure 3 shows samples of play strategies adopted for each style, where the *Aggressive* agent exploits paddle rotation to bounce the ball off the side-wall to outmaneuver the opponent. Conversely, *Defensive* agents attempt to return the ball directly to the opponent, exploiting the paddle rotation to angle the ball back towards the opponent. As desired, aggressive play yields high win-rates, with plays lasting significantly shorter than defensive play, which would win approximately half of the games but play longer games (videos in Supplementary Material).

Table 1: Learning different play-styles on Pong with rotatable player paddles

| | Agent | Single-style | MT w/ one-hot | MN-MultiCriticAL | MH-MultiCriticAL |
|---|---|---|---|---|---|
| | | *Aggressive-Setting → Style = (1, 0)* | | | |
| **SAC** | Reward ↑ | $0.89 \pm 0.05$ | $-0.20 \pm 0.35$ | $0.85 \pm 0.01$ | $0.80 \pm 0.03$ |
| | Win-rate ↑ | $94 \pm 2\%$ | $40 \pm 13\%$ | $86 \pm 3\%$ | $84 \pm 8\%$ |
| | Play-time ↓ | $280 \pm 102$ | $170 \pm 51$ | $255 \pm 35$ | $291 \pm 55$ |
| | | *Defensive-Setting → Style = (0, 1)* | | | |
| | Reward ↑ | $162.75 \pm 12.15$ | $166.05 \pm 13.04$ | $176.60 \pm 8.50$ | $165.59 \pm 13.01$ |
| | Win-rate | $51 \pm 3\%$ | $50 \pm 5\%$ | $30 \pm 5\%$ | $41 \pm 12\%$ |
| | Play-time ↑ | $1536 \pm 113$ | $1571 \pm 122$ | $1661 \pm 53$ | $1560 \pm 154$ |
| | | *Aggressive-Setting → Style = (1, 0)* | | | |
| **PPO** | Reward ↑ | $0.70 \pm 0.12$ | $-0.35 \pm 0.36$ | $0.83 \pm 0.08$ | $0.70 \pm 0.19$ |
| | Win-rate ↑ | $81 \pm 10\%$ | $32 \pm 23\%$ | $79 \pm 2\%$ | $80 \pm 15\%$ |
| | Play-time ↓ | $586 \pm 98$ | $241 \pm 111$ | $485 \pm 44$ | $330 \pm 98$ |
| | | *Defensive-Setting → Style = (0, 1)* | | | |
| | Reward ↑ | $101.35 \pm 17.90$ | $35.85 \pm 16.08$ | $92.75 \pm 12.39$ | $72.60 \pm 14.06$ |
| | Win-rate | $54 \pm 7\%$ | $41 \pm 3\%$ | $40 \pm 3\%$ | $61 \pm 8\%$ |
| | Play-time ↑ | $993 \pm 120$ | $347 \pm 132$ | $841 \pm 226$ | $726 \pm 121$ |

Table 2: Interpolating between aggressive and defensive Pong play-styles → Style = (0.5, 0.5)

| Agent | SAC | | PPO | |
|---|---|---|---|---|
| | MN-MultiCriticAL | MH-MultiCriticAL | MN-MultiCriticAL | MH-MultiCriticAL |
| | *Training with Binary Style Selection* | | | |
| Win-rate | $71 \pm 3\%$ | $64 \pm 13\%$ | $64 \pm 4\%$ | $78 \pm 5\%$ |
| Play-time | $770 \pm 148$ | $765 \pm 113$ | $615 \pm 164$ | $541 \pm 67$ |
| | *Training with Explicit Style Interpolation* | | | |
| Win-rate | $73 \pm 6\%$ | $72 \pm 8\%$ | $71 \pm 3\%$ | $70 \pm 2\%$ |
| Play-time | $834 \pm 108$ | $793 \pm 120$ | $621 \pm 158$ | $602 \pm 106$ |

Multi-style performance is tested by introducing a one-hot style encoder and randomly uniformly sampling either defensive or aggressive play for each new training episode. We observed that both MTSAC and MTPPO failed to learn to properly distinguish between play-styles. MTSAC agents often defaulted to highly defensive play (likely due to the over-representation of the dense defensive rewards in the replay buffers) while MTPPO agents tended to learn intermediate behavior that was neither aggressive nor defensive. MultiCriticAL agents were however able to successfully learn to distinguish between the styles with both base algorithms, with performance comparable to single-style agents. Table 1 and Figure A.2.1, show performance statistics averaged over 20 test episodes per seed for each algorithm.

We also test how MultiCriticAL agents would respond to changes in the style encoding at test time. As shown in Table 2, when provided an intermediate (0.5,0.5) encoding, the agents presented with behavior statistics that were almost halfway between that of their aggressive and defensive statistics. This was even without explicit conditioning, when agents were trained with purely binary style indicators. When explicitly interpolating between styles in training (by providing a multi-dimensional style-reward, with aggressive and defensive rewards scaled by their encoding weight), both MultiCriticAL SAC and PPO agents yielded performance statistics that more closely matched the mid-points between their aggressive and defensive behaviors.

## 5.3 SONIC THE HEDGEHOG

**Training Setup** To test MultiCriticAL in a more established multi-task setting, we tested two Sonic the Hedgehog games, using the Gym-retro wrappers designed for the games during the Gym-retro contest (Nichol et al., 2018), using the same setup configuration developed for the competition. MultiCriticAL is set up to maintain a separate critic for each trained level. Sufficient experimentation for statistical rigor on this benchmark required hardware parallelism, which our code only supports for PPO. SAC is therefore omitted for this benchmark.

**Results** Performance on 17 of the main levels for each game, averaged over 10 tests per agent, per level, are presented in Figure A.2.2, with per-level tabulated numerical rewards provided in Appendix A. Sonic the Hedgehog appears to be a game where learning shared representations over multiple levels helps overall performance, for both MultiCriticAL and MTPPO, though this observation does not hold for Sonic the Hedgehog 2, where MTPPO agents perform worse on average than agents trained on single-levels. MultiCriticAL agents tend to perform comparable or better than independently trained and MTPPO agents in almost all cases, however, with 35% and 20% improvements on Sonic the Hedgehog and 56% and 40% improvement on Sonic the Hedgehog 2 of MN-MultiCriticAL and MH-MultiCriticAL over MTPPO respectively.

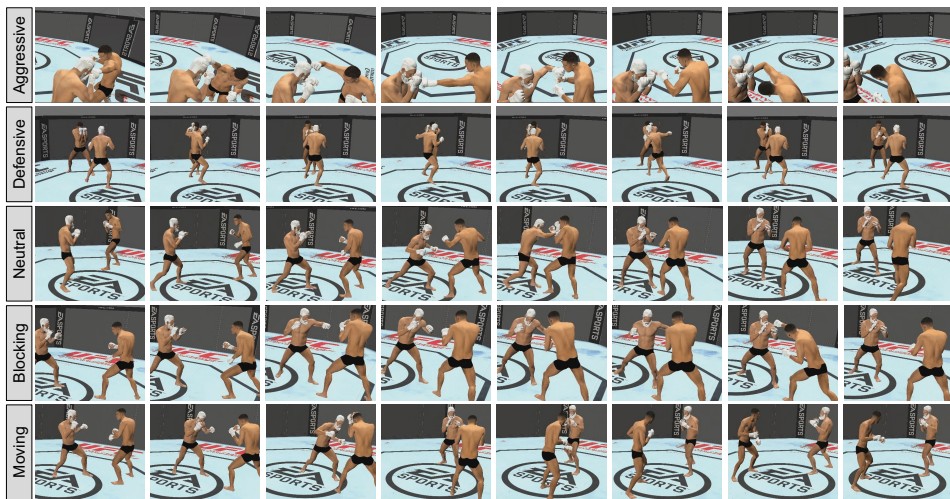

Figure 4: Evaluated over two Sonic games with 17 training levels per game, where levels function similarly but have significantly different layouts and visual design, MultiCriticAL offers consistently superior performance to the single-critic baseline and achieves comparable, if not better performance to agents trained specifically for each level – additional data and visualization in Appendix A.2.2.

Figure 5: We show 8 seconds of play with one representative frame per second for the different styles of play for the MultiCriticAL agent trained to play UFC. The RL agent has black hair while the stock AI has white hair. All styles are captured for the same RL agent, trained to play with multiple behavior styles. As desired, the agent more frequently goes on the offensive when set to *Aggressively*, prioritizes constant blocking and avoiding damage on *Defensive*, standing its ground and blocking incoming attacks on *Blocking* and moving around the ring for *Moving*.

## 5.4 UFC Environment

For a use case more reflective of real-world application, we trained AI fighters in an experimental build of UFC Knockout Mode. A gym-like (Brockman et al., 2016) wrapper was built for the game environment to enable an RL agent to play against the game's stock AI. Note that this is not a feature of the commercially released game but is a modification made for research purposes.

**Environment design and Reward Shaping**    The RL agent observes information about the player's and opponent's position, velocity, health, and actions being taken, stacked over 8 steps. The RL-agent is able to control 11 basic discrete high-level actions including forward and backward movement, high or low blocking, left or right attacking (with straights, uppercuts, or hooks), and is also allowed to take no action. The RL-agent is conditioned over 2 sets of 2 playing styles: aggressive or defensive and blocking or moving. During training, aggression vs. defensiveness, and blocking vs. moving are set on $[0, 1]$ scales with $(0.5, 0.5)$ representing a neutral style which is meant to balance behavior from both scales. For all styles, the agent is rewarded for winning and is penalized for every strike request. For the aggressive style, the agent is additionally rewarded for reducing opponent health. For the defensive style, the agent gets a small positive reward for each step that it remains alive, is penalized for losing health, and is rewarded for reducing opponent health (though not as much as with the aggressive style). For the blocking style, the agent is rewarded for every successful block, and for the moving style, the agent is rewarded for maintaining a positive velocity. Details are provided on the state and action spaces, reward tuning, and training in Appendix D.4.

Table 3: MTPPO vs. Multi-headed MultiCriticAL-PPO on UFC

| Dominant Style | Style Setting | Normalized Per-Action Style Rewards $\times 10^3$ | | | | | | | |
|---|---|---|---|---|---|---|---|---|---|
| | | MTPPO | | | | MH-MultiCriticAL PPO | | | |
| | | Aggression | Defensiveness | Blocking | Moving | Aggression | Defensiveness | Blocking | Moving |
| Aggressive | (0.0, 0.5) | $7.08 \pm 1.04$ | 0.00 | $0.10 \pm 0.05$ | $1.77 \pm 0.15$ | $\mathbf{7.15 \pm 1.51}$ | 0.00 | $0.05 \pm 0.02$ | $2.19 \pm 0.21$ |
| Defensive | (1.0, 0.5) | 0.00 | $20.08 \pm 0.30$ | $\mathbf{4.31 \pm 0.41}$ | $1.19 \pm 0.12$ | 0.00 | $\mathbf{20.80 \pm 0.07}$ | $4.27 \pm 0.37$ | $2.33 \pm 0.17$ |
| Neutral | (0.5, 0.5) | $2.84 \pm 0.90$ | $14.15 \pm 0.74$ | $1.32 \pm 0.50$ | $1.93 \pm 0.07$ | $2.16 \pm 0.15$ | $10.80 \pm 0.12$ | $1.68 \pm 1.68$ | $2.24 \pm 0.15$ |
| Blocking | (0.5, 0.0) | $2.25 \pm 1.25$ | $18.86 \pm 1.07$ | $3.10 \pm 3.78.$ | 0.00 | $3.00 \pm 1.14$ | $11.56 \pm 1.32$ | $1.97 \pm 2.32$ | 0.00 |
| Moving | (0.5, 1.0) | $3.07 \pm 1.01$ | $14.78 \pm 1.13$ | 0.00 | $4.19 \pm 0.43$ | $1.75 \pm 1.25$ | $10.13 \pm 0.88$ | 0.00 | $\mathbf{4.50 \pm 0.20}$ |

**Results** MultiCriticAL allows PPO agents to successfully learn and smoothly transition between the different fighting styles. The behavior styles are distinguished visually and Figure 5 attempts to capture this through frames from captured gameplay with videos included in the Supplementary Material. Reward statistics for each style are presented in Table 3 and Figure A.2.3. Aggressive and defensive plays are the most distinct but there are also observable differences between control that prioritizes blocking and moving, with the blocking-style conditioning agents to hold their ground while moving prompts agents to move around the ring more. When comparing MultiCriticAL to MTPPO, we note that, while rewards at the extremes of style distinction are similar, MultiCriticAL appears to be better at more consistently maintaining intermediate styles (when a setting is at 0.5). Visually, it appears that MTPPO also tended to favor blocking and defense over learning more nuanced behavior (Table 6).

## 5.5 DISCUSSION

While explicit separation of the learned value functions per style under the MultiCriticAL framework offers a demonstrable and consistent improvement over the single-critic baselines, it must be noted that MTSAC and MTPPO baselines do not necessarily reflect the state-of-the-art in MTRL, but they do reflect how value optimization is typically handled in contemporary literature. It is likely that utilizing some of the improvements to policy optimization techniques discussed in Section 3, may contribute to improved performance of policies trained using single-critic value-optimization, however the same may also further improve the performance of agents trained with MultiCriticAL and remains to be studied. Curiously, MN-MultiCriticAL consistently outperformed MH-MultiCriticAL on all the tested tasks, which calls into question the idea that shared representation is important and/or useful (Dewangan et al., 2018; Yang et al., 2017) and is an aspect that may warrant further study. An interesting extension could be to extend modularization (Yang et al., 2020; Andreas et al., 2017) to value-function optimization. Learning with curricula (Bengio et al., 2009; Graves et al., 2017; Mysore et al., 2019; Andreas et al., 2017) has also been demonstrated to (sometimes dramatically) improve RL performance and multi-style/task curricula also seem like a potential natural fit for multi-critic optimization. Recent work (Mysore et al., 2021a; Andrychowicz et al., 2021; Ilyas et al., 2020) has also demonstrated that value function capacity can be a key bottleneck in RL performance. Increasing the network capacity of a single-critic value-function may improve the efficacy of the critic, but this is not guaranteed and there are few principled ways to determine what a sufficient network capacity may be. Significantly increased computational resources may be required for the learnability of the multi-style problems under a single-critc framework. MultiCriticAL also requires increased compute during training, particularly MN-MultiCriticAL, but MH-MultiCriticAL allows for a bounded increase to computational load ($< 3\%$ as tested - see Appendix E) with an acceptable compromise in relative performance while still outperforming single-critic value learning.

## 6 CONCLUSION

We study Actor-Critic optimization in multi-style RL problems, where agents are required to behave with different styles in the same environments, our work represents one of the first explorations of the application of RL to a wider array of multi-style problems. We hypothesized that limitations of single-actor, single-critic MTRL frameworks may result in interference between the reward signals for different styles due to high similarity in visited states and to mitigate this, we proposed Multi-Critic Actor Learning (MultiCriticAL), which separates critics into per-task value-function estimators, thus avoiding interference between style rewards and reduces learning complexity by not requiring critics to learn to distinguish between tasks. MultiCriticAL consistently outperforms the single-critic MTRL baselines, achieving up to 56% improvement in multi-task performance and successfully learning multiple behavior styles where single-critic methods would fail.

## REPRODUCIBILITY STATEMENT

As noted in Section 5, training code is based upon the open-source OpenAI Spinning Up code-base (Achiam, 2018), with modifications made to support MultiCriticAL for PPO and SAC. The modified code-base is included in the Supplementary Material for this paper, along with the code for the custom environments for Path Following (presented in Section 5.1) and Pong with rotatable paddles (presented in Section 5.2). The README files included with the code detail installation and use of the code. For the Sonic the Hedgehog games, reproducing results would require acquiring the games' ROM files and linking them to an install of gym-retro, as detailed by Nichol et al. (2018). Also included in Appendices D and E are additional details for the setup of the training environments discussed in the main body of the paper as well as training configurations used. We recognize that tests on the UFC environment may not be independently reproducible due to requiring access to proprietary development software and we therefore mainly discuss it as a demonstration of the practical utility for our proposed MultiCriticAL method, rather than as a benchmark task.

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

# A  ADDITIONAL DATA AND RESULTS

## A.1  LEARNING CURVES

### A.1.1  PATH FOLLOWING

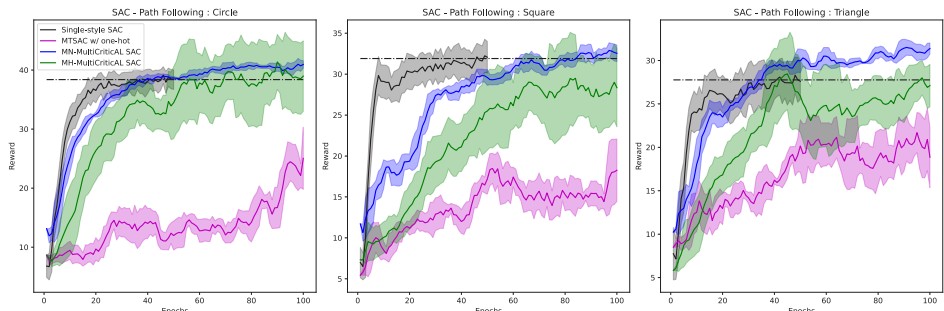

Figure 6: Learning Curves for all tested SAC variants on the Path Following benchmark.

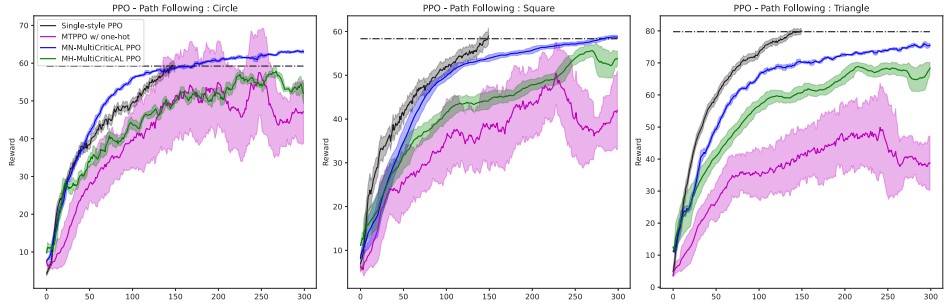

Figure 7: Learning Curves for all tested PPO variants on the Path Following benchmark.

### A.1.2  PONG

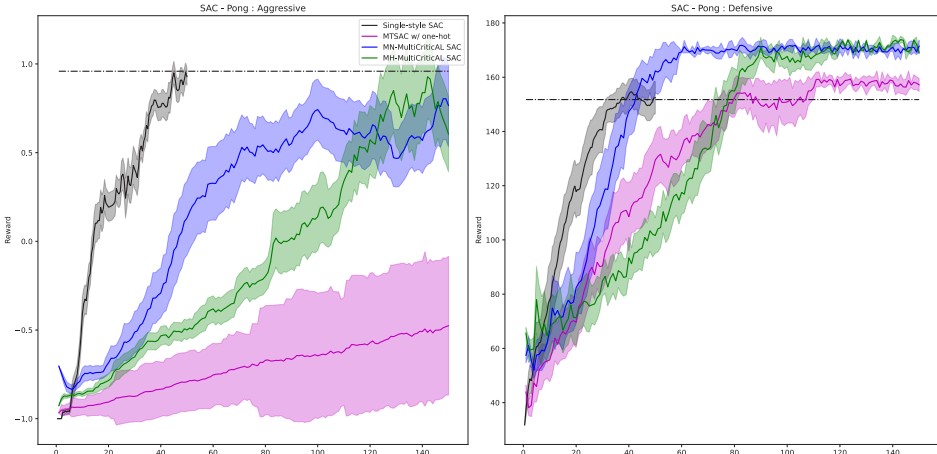

Figure 8: Learning Curves for all tested SAC variants on the Pong benchmark.

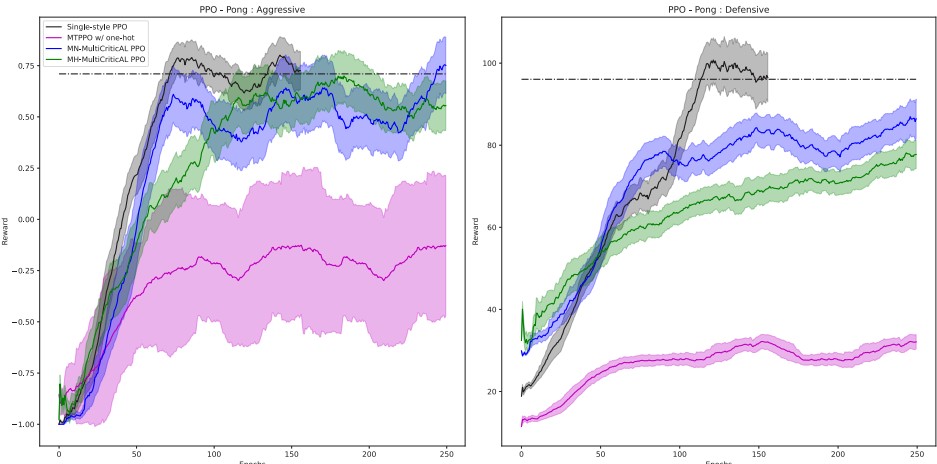

Figure 9: Learning Curves for all tested PPO variants on the Pong benchmark.

### A.1.3 SONIC THE HEDGEHOG

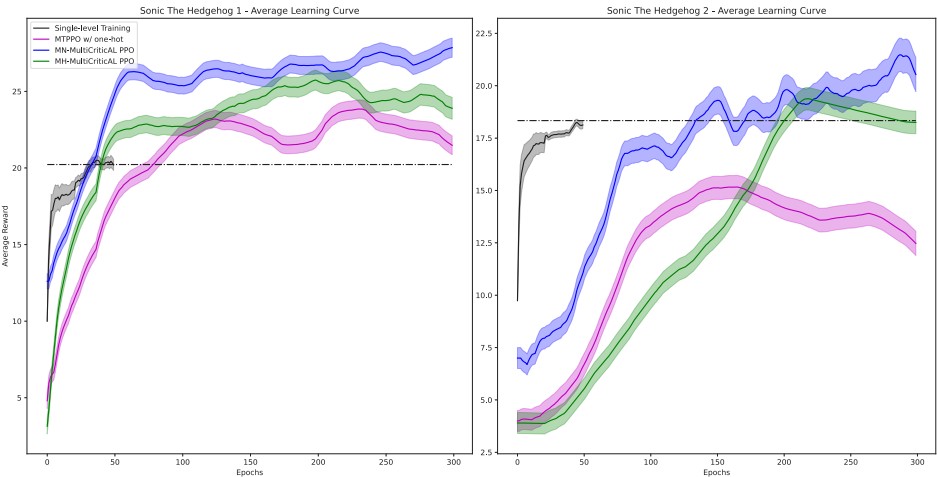

Figure 10: Average Learning Curves the Sonic the Hedgehog benchmark games. Represented here is the averaged reward over the 17 levels of the game as training progresses. We chose this representation in the interest of providing a more compact representation instead of including 34 separate figures, with one for each level, as that would both require a lot of space and be more difficult to parse.

## A.2 ALTERNATIVE RESULTS REPRESENTATION

### A.2.1 PONG

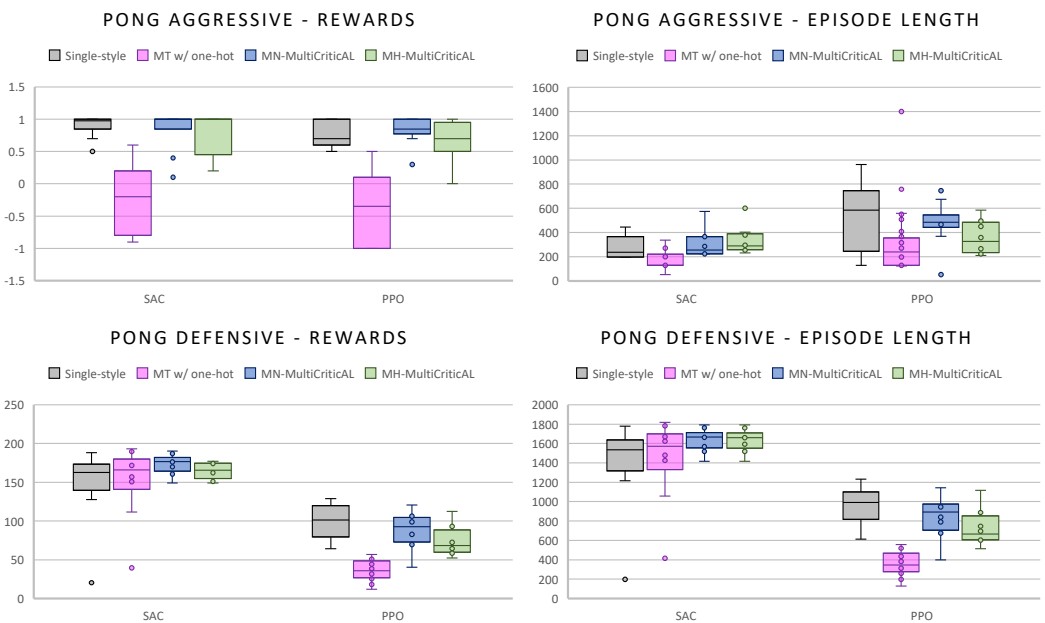

Figure 11: Box and Whisker plots for Pong Results presented in Section 5.

### A.2.2 SONIC THE HEDGEHOG

Table 4: Rewards achieved on Sonic the Hedgehog (1)

| Abbreviation | Level | Single-level Policy | MTPPO | MN-MultiCriticAL | MH-MultiCriticAL |
|---|---|---|---|---|---|
| GHZ1 | GreenHillZone.Act1 | $37.41 \pm 15.84$ | $38.70 \pm 23.41$ | $\mathbf{42.30 \pm 5.32}$ | $40.72 \pm 6.04$ |
| GHZ2 | GreenHillZone.Act2 | $17.60 \pm 9.66$ | $26.40 \pm 10.62$ | $\mathbf{42.26 \pm 2.07}$ | $32.63 \pm 6.01$ |
| GHZ3 | GreenHillZone.Act3 | $7.22 \pm 4.07$ | $9.37 \pm 6.58$ | $\mathbf{38.85 \pm 25.18}$ | $12.02 \pm 5.54$ |
| MZ1 | MarbleZone.Act1 | $28.03 \pm 5.45$ | $26.20 \pm 11.30$ | $39.93 \pm 13.30$ | $\mathbf{40.40 \pm 8.45}$ |
| MZ2 | MarbleZone.Act2 | $18.86 \pm 5.23$ | $16.79 \pm 6.60$ | $22.84 \pm 1.70$ | $\mathbf{22.92 \pm 0.11}$ |
| MZ3 | MarbleZone.Act3 | $27.31 \pm 2.36$ | $22.64 \pm 7.04$ | $\mathbf{29.82 \pm 0.68}$ | $28.50 \pm 0.17$ |
| SYZ1 | SpringYardZone.Act1 | $6.29 \pm 2.73$ | $7.82 \pm 6.50$ | $\mathbf{10.02 \pm 5.34}$ | $9.16 \pm 5.14$ |
| SYZ2 | SpringYardZone.Act2 | $5.55 \pm 3.71$ | $11.59 \pm 2.85$ | $\mathbf{18.65 \pm 3.41}$ | $13.47 \pm 6.06$ |
| SYZ3 | SpringYardZone.Act3 | $19.67 \pm 2.19$ | $17.17 \pm 1.58$ | $22.82 \pm 4.97$ | $\mathbf{26.14 \pm 2.37}$ |
| LZ1 | LabyrinthZone.Act1 | $22.50 \pm 3.18$ | $22.04 \pm 5.00$ | $22.00 \pm 6.40$ | $\mathbf{25.16 \pm 12.77}$ |
| LZ2 | LabyrinthZone.Act2 | $29.52 \pm 0.40$ | $\mathbf{29.65 \pm 0.19}$ | $29.59 \pm 0.31$ | $29.46 \pm 0.06$ |
| LZ3 | LabyrinthZone.Act3 | $25.70 \pm 4.02$ | $26.04 \pm 0.09$ | $\mathbf{26.33 \pm 0.13}$ | $25.82 \pm 0.19$ |
| SLZ1 | StarLightZone.Act1 | $35.03 \pm 0.51$ | $35.15 \pm 1.06$ | $\mathbf{47.80 \pm 12.38}$ | $39.79 \pm 4.38$ |
| SLZ2 | StarLightZone.Act2 | $6.76 \pm 0.14$ | $13.05 \pm 4.62$ | $\mathbf{18.25 \pm 0.30}$ | $17.04 \pm 3.21$ |
| SLZ3 | StarLightZone.Act3 | $14.47 \pm 6.95$ | $26.37 \pm 5.01$ | $\mathbf{30.40 \pm 6.39}$ | $30.20 \pm 1.00$ |
| SBZ1 | ScrapBrainZone.Act1 | $8.03 \pm 1.65$ | $10.73 \pm 4.52$ | $9.92 \pm 4.13$ | $\mathbf{11.68 \pm 6.11}$ |
| SBZ2 | ScrapBrainZone.Act2 | $12.52 \pm 0.00$ | $8.54 \pm 4.27$ | $\mathbf{11.61 \pm 1.99}$ | $11.25 \pm 1.51$ |
| Average | | $19.42 \pm 2.03$ | $20.48 \pm 2.24$ | $\mathbf{27.39 \pm 2.65}$ | $24.73 \pm 1.31$ |

Table 5: Rewards achieved on Sonic the Hedgehog 2

| Abbreviation | Level | Single-level Policy | MTPPO | MN-MultiCriticAL | MH-MultiCriticAL |
|---|---|---|---|---|---|
| EHZ1 | EmeraldHillZone.Act1 | 48.98 ± 10.01 | 43.72 ± 17.08 | **65.82 ± 15.48** | 44.03 ± 6.07 |
| EHZ2 | EmeraldHillZone.Act2 | 21.82 ± 2.37 | 18.43 ± 10.09 | **28.31 ± 6.27** | 23.65 ± 13.28 |
| CPZ1 | ChemicalPlantZone.Act1 | 26.58 ± 3.29 | 28.21 ± 2.01 | **37.48 ± 10.18** | 33.65 ± 11.64 |
| CPZ2 | ChemicalPlantZone.Act2 | 28.82 ± 4.32 | 18.84 ± 8.90 | 24.60 ± 9.85 | **27.01 ± 5.00** |
| ARZ1 | AquaticRuinZone.Act1 | 15.30 ± 7.31 | 5.85 ± 0.00 | **24.28 ± 5.56** | 19.18 ± 9.24 |
| ARZ2 | AquaticRuinZone.Act2 | 37.40 ± 6.89 | 24.12 ± 2.85 | **39.41 ± 7.30** | 33.15 ± 10.23 |
| CNZ | CasinoNightZone.Act1 | 17.12 ± 0.97 | 12.02 ± 3.51 | **20.34 ± 5.01** | 15.65 ± 1.99 |
| HTZ1 | HillTopZone.Act1 | 7.82 ± 0.20 | 6.36 ± 2.72 | **7.76 ± 0.85** | 5.65 ± 0.57 |
| HTZ2 | HillTopZone.Act2 | 6.44 ± 4.23 | 4.22 ± 2.81 | 19.89 ± 2.96 | **21.30 ± 3.08** |
| MCZ1 | MysticCaveZone.Act1 | 10.25 ± 0.34 | 7.42 ± 1.14 | 9.74 ± 1.44 | **10.23 ± 0.17** |
| MCZ2 | MysticCaveZone.Act2 | 8.57 ± 2.31 | 5.47 ± 0.00 | 8.75 ± 3.32 | **11.96 ± 1.59** |
| OOZ1 | OilOceanZone.Act1 | 14.95 ± 3.87 | 8.65 ± 4.18 | **19.14 ± 2.58** | 13.04 ± 4.28 |
| OOZ2 | OilOceanZone.Act2 | 10.20 ± 0.90 | 9.92 ± 2.11 | 13.72 ± 4.16 | **14.06 ± 3.22** |
| MZ1 | MetropolisZone.Act1 | 14.64 ± 3.25 | 6.07 ± 1.94 | **14.39 ± 2.58** | 14.03 ± 3.52 |
| MZ2 | MetropolisZone.Act2 | 12.79 ± 2.68 | 11.50 ± 2.53 | 13.12 ± 0.37 | **16.90 ± 7.06** |
| MZ3 | MetropolisZone.Act3 | 6.49 ± 2.27 | 10.31 ± 1.59 | 11.75 ± 2.29 | **17.34 ± 5.16** |
| WFZ | WingFortressZone | 27.43 ± 0.88 | 8.74 ± 0.41 | **24.81 ± 3.98** | 21.29 ± 5.53 |
| Average | | 18.90 ± 1.71 | 13.73 ± 1.06 | **21.11 ± 2.44** | 19.22 ± 1.95 |

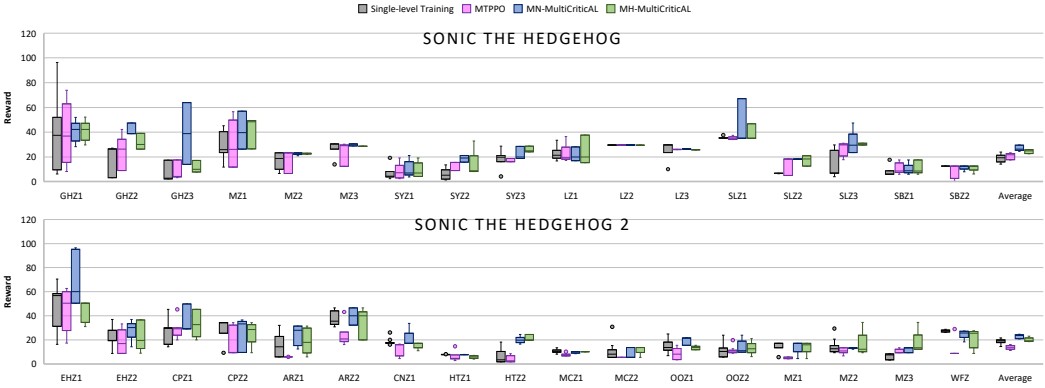

Figure 12: Sonic Results from Tables 4 and 5 visualized

### A.2.3 UFC REWARD AND ACTION STATISTICS

Table 6: Action probabilities for MTPPO and MH-MultiCriticAL-PPO on UFC

| Dominant Style | No-action | Strike | Block | Move | Win Rate (%) |
|---|---|---|---|---|---|
| MTPPO | | | | | |
| Aggressive | 3.80 | **32.99** | 3.60 | **59.61** | 96 |
| Defensive | 7.48 | 5.42 | **73.86** | 13.22 | 4 (48% ties) |
| Neutral | 7.48 | 20.97 | 24.12 | 47.42 | **100** |
| Blocking | **8.31** | 20.89 | 27.21 | 43.59 | 96 |
| Moving | 6.46 | 26.75 | 15.91 | 50.89 | 88 |
| MH-MultiCriticAL-PPO | | | | | |
| Aggressive | 0.27 | **15.04** | 15.78 | 68.91 | **96** |
| Defensive | 0.52 | 4.94 | **30.20** | 64.34 | 12 (56% ties) |
| Neutral | **0.57** | 11.30 | 22.88 | 65.25 | 88 (4% ties) |
| Blocking | 0.49 | 14.88 | 23.92 | 60.71 | 88 |
| Moving | 0.59 | 8.70 | 20.88 | **69.82** | 64 (8% ties) |

We observe from Figure A.2.3 and Table 6 that MultiCriticAL more consistently learns delineated styles that are consistent with the behaviors that the rewards try to encourage. Note that, with our method, it is, in fact, the aggressive style that wins most fights consistently and the defensive style learns to block and move to not lose as quickly, whereas MTPPO tends seems to equate defensive play with just blocking, and this also contributes to a higher rate of losing. We see as well that the intermediate settings correspond to a more balanced blend of action probabilities between extremes when compared to MTPPO.

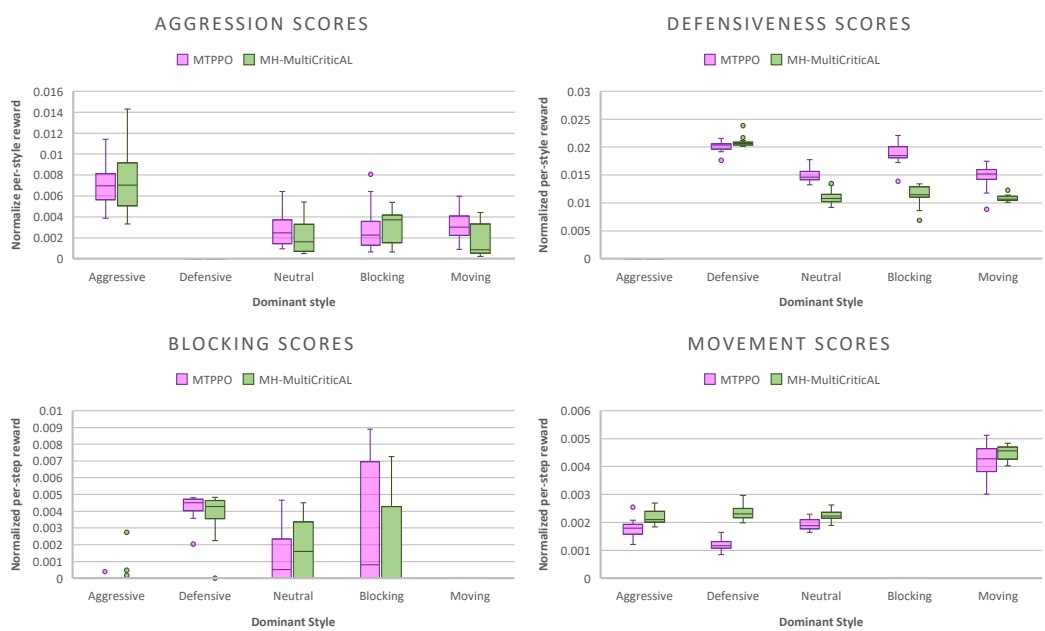

Figure 13: Box and Whisker plots for UFC style rewards presented in Section 5.

## A.3 PONG ACTION INTERPOLATION WITH MULTI-POLICY COMPOSITION

As an exercise in curiosity, we also consider what would happen if we attempted to derive intermediate styles by interpolating between actions in the Pong game. This was meant to test an approach similar to the multi-policy action composition technique used by Peng et al. (2018). Unlike in their case where actions were weighted according to each actor's estimated critic value, we instead directly interpolate between actions proposed by pre-trained aggressive and defensive agents. We take this approach as the training for either style would not offer the value functions any meaningful information about the other unseen style(s), prohibiting the form of value comparison employed by Peng et al. We hypothesized that it is possible and even likely that policies would destructively interfere with each other if the longer-term goals of constituent policies conflict with each other and this is exactly what was observed in testing. Except for intermediate styles very close to the learned extremes, interpolating between actions proposed by the aggressive and defensive agents caused a near-complete failure in the composite policy's control.

Composite policies were built by pairing a randomly selected pre-trained aggressive policy with a randomly selected defensive policy. On a (0.5,0.5) 'middle' setting, SAC composite policies achieved an average win-rate of $30\% \pm 5\%$ (compared to the approximately 95% and 50% of the constituent policies) with an average Play-time of $213 \pm 89$ steps (often losing the game quickly). Similarly, composite PPO also achieves a low average win-rate of $15\% \pm 3\%$ and similarly lose the game quickly. A high win-rate would not be expected, as the defensive policies are explicitly trained to avoid winning, but both policy classes are trained to avoid losing. Given the high loss-rate, composing policies in this way, for this multi-style task, can only be regarded as a failure. By contrast however, as we showed in Table 2, MultiCriticAL is capable of learning to interpolate between styles, both when and not explicitly trained to do so.

This experiment, by no means, constitutes exhaustive testing, and cannot be used to claim that all, or even most, forms of multi-policy policy composition would be ineffective and is therefore not included in the main body of the paper. We do believe however that it helps highlight a circumstance where policy composition can fail in cases where long-term strategies for different target behavior styles may interfere. Considered more generally, there are no guarantees that actions proposed by distinct policies would be similar enough, that a weighted sum over them would even make sense, as cooperation between policies is not something that is explicitly trained for. One might benefit from explicit conditioning, though this was outside the scope of our work.

## B  RELATING MULTICRITICAL TO DQNS AND MULIT-Q LEARNING

The idea of using multiple Q functions to guide the training of a single policy is not strictly new to the broader context of RL and has been used in notably single-task RL for improving the stability of learned value functions. These include techniques such as Double Q-learning (Hasselt, 2010; Van Hasselt et al., 2016), Dueling Q-networks (Wang et al., 2016), and Bootstrapped DQNs (Osband et al., 2016; Steckelmacher et al., 2019). There are also applications in multi-task learning, though we only encountered them in works considering DQN frameworks that, in effect, learn task-specific Q values for a single policy in DQN architectures (Rusu et al., 2016; 2015; D'Eramo et al., 2020). As DQNs jointly encode both the action policies and the value representation in the same network, however, they do not share the practical benefits offered by Actor-Critic methods (and consequently MultiCriticAL) in separating the actors and critics. We believe that, practically, this is a meaningful distinction as DQN methods are (i) limited to discrete action spaces and (ii) need the full representational and computational complexity of the value estimation to be preserved at inference time after training, which is computationally inefficient. As shown by Andrychowicz et al. (2021) and Mysore et al. (2021a), oftentimes, it appears that the value function often bears the higher burden of required complexity to enable successful policy learning. Noting this, there is a significant practical advantage to being able to just train smaller, less computationally expensive actor-networks of actor-critic methods and disregard the value estimation networks at inference time. Given that our work is largely motivated by application to video games, where runtime resource use is an important consideration, this was an important distinction.

## C  MULTI-NETWORK VS. MULTI-HEADED TRAINING FRAMEWORK OVERVIEW

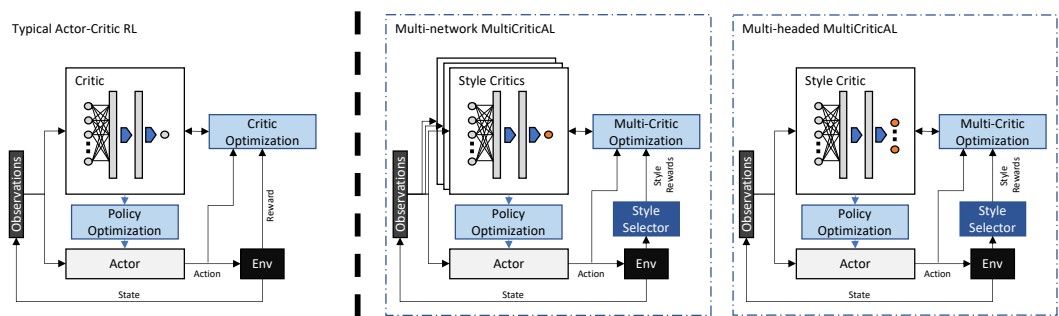

Figure 14:  MN-MultiCriticAL and MH-MultiCriticAL frameworks compared. The main differences between the multi-network (MN) and multi-headed (MH) is in how the critic functions are ultimately represented by the neural network(s). In the MN case, separate and independent networks are maintained for each style/task value function, whereas the MH configuration allows value functions to share a common backbone and instead utilizes different network heads (i.e. output nodes) for each learned value. While the former appears to offer superior performance in most of the tested cases, the latter can be less computationally expensive and may incur reduced training costs. It should be noted, however, that neither architecture needs to impact the actor, thus preserving run-time inference cost.

# D  ADDITIONAL ENVIRONMENT SETUP DETAILS

## D.1  PATH FOLLOWING

**Observed State**  Agents observe a 4-dimensional state with the current $(x, y)$ coordinates of the agent in space and the current phase $\rho \in [0, 1]$ represented by $[\sin(2\rho\pi), \cos(2\rho\pi)]$

**Goal generation**  Goals, $g$ are generated based on the current phase of motion. They are as follows:

- Circle: $g(\rho) = [\sin(2\rho\pi), \cos(2\rho\pi)] \ \ \forall \rho \in [0, 1]$

- Square: $g(\rho) = \begin{cases} [4\rho - 0.5, 0.5] & \rho \leq 0.25 \\ [0.5, 1.5 - 4\rho] & 0.25 \leq \rho \leq 0.5 \\ [2.5 - 4\rho, -0.5] & 0.5 \leq \rho \leq 0.75 \\ [-0.5, 4\rho - 3.5] & 0.75 \leq \rho \leq 1 \end{cases}$

- Triangle: $g(\rho) = \begin{cases} [1.5\rho, 0.5 - 3\rho] & \rho \leq 1/3 \\ [1.5 - 3\rho, -0.5] & 1/3 \leq \rho \leq 2/3 \\ [-1.5 + 1.5\rho, 1.5\rho - 1.5] & 2/3 \leq \rho \leq 1 \end{cases}$

**Reward design**  For current state $s$ and desired state $g$: $r = 1 - \log(|s - g|) / \log(\max(\Delta s))$ where $\max(\Delta s)$ is the difference between the extents of the space bounds.

## D.2  PONG

**Observed State**  Agents observe an 8-dimensional state with the current vertical $y$ positions of both the player and opponent, the player's velocity, the paddle angle, and the ball's $(x, y)$ position and velocity.

**Reward design**  Aggressive play is conditioned with a sparse reward: +1 on a win and -1 on loss. Agents are meant to learn that the best way to maximize cumulative discounted rewards is to win as quickly as possible. Defensive play is conditioned with dense rewards: -1 on loss, -1 for bouncing the ball against walls, +0.5 for receiving the ball, +0.1 if the opponent successfully receives the ball, and +0.1 for every time-step of play. Defensive rewards encourage avoiding losing and ensuring that the opponent can receive the ball in order to prolong the game.

## D.3  SONIC THE HEDGEHOG

**Reward scaling**  As recommended by Nichol et al. (2018), we scale rewards by a factor of 0.01 in order to encourage more stable learning with PPO.

## D.4  EA UFC DEVELOPMENT ENVIRONMENT

**Environment**  The UFC environment is an experimental build of UFC. During training simulation, superfluous elements of the game, such as audio, spectators, etc. are disabled. Socket communication is used to interface between the game-code and the python-based RL algorithms. The environment wrapper controls the game simulation and extracts and processes state information before communicating it through the socket to the python RL code and interprets requests from the python code to trigger inputs in the game. During training, agents are trained for 240 epochs with 4000 sample interactions collected per epoch. Training is run in increments of 30 epochs to improve simulation and memory management stability. Episodes are terminated when the agent runs out the max episode length or if the health differential between the agent and opponent is over 40 units.

**State and Action Space for the UFC environment**   The environment returns the following data as 30-dimensional observations for the RL agent:

- Position: (X,Z) coordinates in the ring
- Velocity: scalar of trajectory velocity
- Health: [0,100] float value
- Action active: high-level action indicator
- Strike type: indicator of type of strike performed
- Strike segment: indicates the current stage of striking
- Ticks to event: time until next event in game ticks
- Target hit location: the region of the opponent's body the agent is targeting (upper or lower)
- Block successful: indicates when a strike is successfully blocked

The agent is allowed to take the following actions:

- No action
- (Strike) Straight: left or right, single request
- (Strike) Uppercut: left or right, single request
- (Strike) Hook: left or right, single request
- Block: high or low, needs to be held to sustain blocking
- Move: forward or backward relative to current opponent position

**Reward Design**   The reward breakdown for the environment is as follows:

- Standard to all styles – Victory reward: +1; Strike request penalty: -0.001
- Aggressive style – successful attack reward: $+|\Delta\text{Health}_{\text{opponent}}|/100$
- Defensive style – survival reward: +0.02 per step; attack reward: $+|\Delta\text{Health}_{\text{opponent}}|/200$, health loss penalty: $-\Delta\text{Health}_{\text{agent}}/50$
- Blocking style – block reward: +0.01 per successful block
- Movement style – velocity reward: $+0.05v$

# E   TRAINING CONFIGURATION

Training code is based on OpenAI's Spinning Up (Achiam, 2018) and is provided in the zip folder included as additional supplementary material with our main paper submission.

With the exception of hidden layer configuration, we use the same default hyperparameters for each of the algorithms used. Different hidden layer configurations were selected for each environment such that they would not be needlessly large but would consistently solve the problem in the single-style cases

**Network Configuration**   Hidden layer configurations for both the actors and critics per environment:

- Path following: PPO [64,64], SAC [8,8]
- Pong: PPO [64,64], SAC [32,32]
- Sonic the Hedgehog 1 and 2: PPO [256,256]. Note additionally that the CNN feature extractor is shared between the actors and all critics in any particular training run, though they are not shared between separate training runs. The CNN feature extractor was configured based on the the extractor used by Mnih et al. (2015) as it is the same configuration used in the Retro-contest (Nichol et al., 2018).

The above network configurations correspond to the following numbers of trainable weights for each of the critic network(s).

Table 7: Trainable Critic parameters per task

| Task | Single-style | MT w/ one-hot | MN-MultiCriticAL | MH-MultiCriticAL | MH-MultiCriticAL vs. MT (%) |
|---|---|---|---|---|---|
| SAC | | | | | |
| Path Following | 137 | 161 | 483 | 163 | ↑ 1.24% |
| Pong | 5061 | 7109 | 14218 | 7164 | ↑ 0.77% |
| PPO | | | | | |
| Path Following | 4545 | 4737 | 14211 | 4867 | ↑ 2.74% |
| Pong | 68353 | 76545 | 153090 | 76802 | ↑ 0.33% |
| Sonic | 197377 | 201729 | 3429393 | 205841 | ↑ 2.04% |

**Misc. details**   Additionally, a frame wrapper is used with Pong and the Sonic games to augment state information with information from the last 4 observed states. This wrapper is based on the FrameStack wrapper provided in OpenAI Baselines (Dhariwal et al., 2017).

Training time was determined by running training until performance plateaued, with an additional buffer given to multi-task training to account for the multi-task problem setting, with initial tests conducted to ensure that performance indeed plateaued.

**Run configurations**   Configurations for training per environment are as follows:

**Path following**

- Single-style PPO – Trained for 100 epochs with 4000 steps per epoch and 90 steps per episode
- Single-style SAC – Trained for 50 epochs with 4000 steps per epoch and 90 steps per episode
- Multi-style MTPPO – Trained for 300 epochs with 4000 steps per epoch and 90 steps per episode
- Multi-style MTSAC – Trained for 150 epochs with 4000 steps per epoch and 90 steps per episode
- Multi-style MultiCriticAL-PPO – Trained for 300 epochs with 4000 steps per epoch and 90 steps per episode
- Multi-style MultiCriticAL-SAC – Trained for 150 epochs with 4000 steps per epoch and 90 steps per episode

**Pong with rotatable paddles**

- Single-style PPO – Trained for 150 epochs with 4000 steps per epoch and variable length episodes
- Single-style SAC – Trained for 50 epochs with 4000 steps per epoch and variable length episodes
- Multi-style MTPPO – Trained for 250 epochs with 4000 steps per epoch and variable length episodes
- Multi-style MTSAC – Trained for 150 epochs with 4000 steps per epoch and variable length episodes
- Multi-style MultiCriticAL-PPO – Trained for 250 epochs with 4000 steps per epoch and variable length episodes
- Multi-style MultiCriticAL-SAC – Trained for 150 epochs with 4000 steps per epoch and variable length episodes

**Sonic the Hedgehog (both 1 and 2)**

- Single-style PPO – Trained for 30 epochs with 10000 steps per epoch and up to 2500 steps per episode, or until Sonic's death, whichever occurs first
- Multi-style MTPPO – Trained for 300 epochs with 10000 steps per epoch and up to 2500 steps per episode, or until Sonic's death, whichever occurs first
- Multi-style MultiCriticAL-PPO – Trained for 300 epochs with 10000 steps per epoch and up to 2500 steps per episode, or until Sonic's death, whichever occurs first

