# OpenReview forum: "Multi-Critic Actor Learning: Teaching RL Policies to Act with Style"
_ICLR.cc/2022/Conference — ICLR 2022 Poster_

### Official Review · Reviewer_Lh1z · 2021-10-19

**Correctness:** 3
**Technical Novelty And Significance:** 3
**Empirical Novelty And Significance:** 3
**Recommendation:** 8
**Confidence:** 4

**Main Review:**

The paper is very well written, and to me, easy to understand and to follow. It is also well-motivated, and the related work section seems quite complete. It seems that the authors did not believe themselves when they found out that single-actor multiple-critics settings are rare in multi-task RL, and spent a considerable amount of time looking for these papers.

Speaking of related work, it cites the Actor-Mimic paper [3], in which N critics are trained on N tasks, and then the actor is distilled to be good at all these tasks. The different with this paper, I think, is that the Actor-Mimic actor does not observe a task descriptor. Am I right? I have the feeling that the Actor-Mimic may deserve a bit more than an entry in a citation list in the background section, as it seems quite close to the main contribution of this paper.

One small remark regarding the experiments is that the authors focus on the impact of multiple critics (which is well-motivated). Our of curiosity, I would have been interested to see at least one experiment that compares MultiCriticAL with the state of the art (actor-based) of multi-task RL, just to see what fancy actors allow to do in comparison with multiple critics.

Regarding the related work section, already very complete, a mention of Bootstrapped DQN could be added (with a mention that it is not multi-task) in case the paper is attacked on the novelty of multiple critics. Multiple critics are common in single-task RL, to provide uncertainty measures and good exploration [1, 2], but I'm not aware of an application to multi-task RL.

Overall, I really like this paper, and recommend accepting it. It identifies an interesting problem and omission in the literature, and proposes a simple solution that performs well in many experiments.

[1]: Osband, I., Blundell, C., Pritzel, A., & Van Roy, B. (2016). Deep exploration via bootstrapped DQN. Advances in neural information processing systems, 29, 4026-4034.
[2]: Steckelmacher, D., Plisnier, H., Roijers, D. M., & Nowé, A. (2019). Sample-Efficient Model-Free Reinforcement Learning with Off-Policy Critics. arXiv preprint arXiv:1903.04193.
[3]: Parisotto, E., Ba, J. L., & Salakhutdinov, R. (2015). Actor-mimic: Deep multitask and transfer reinforcement learning. arXiv preprint arXiv:1511.06342.

**Summary Of The Paper:**

The paper considers the setting in which a single RL policy has to be learned for several tasks (or styles), to be selected at run-time after the agent has been trained on all the tasks. The core of the paper is the identification of a gap in the literature, where it has not been tried to train several critics and a single actor for multi-task RL, with each critic seeing only one task to be learned. The paper proposes to do just that, and shows promising experimental results in a variety of environments.

**Summary Of The Review:**

The paper is well-written, easy to follow, and well-position regarding related work (with a note on the Actor-Mimic). The proposed method is simple, yet unique and performs well. An extra experiment that compares MultiCriticAL with policy-based multi-task approaches would have been nice, but I overall still recommend accepting this paper.

---

> ### Author Response · Authors · 2021-11-12
> **Initial response to Reviewer Lh1z**
>
> We would like to begin by thanking you for your positive review and vote to have our paper accepted for publication. With this initial response, we present our plan to revise the paper within the next few days to address concerns raised about our paper and also try to address a few of the queries you raised about our work. Please let us know if there are elements you would offer further notes on.
>
> Firstly, regarding your note about Actor-Mimic, I believe your observation is largely correct, though given that the actor-mimic networks (AMNs) are actually based on the DQN architecture (see below for a brief explanation of why we make a distinction), it is not quite the same as with actor-critic methods (despite the AMN’s name). DQNs do constitute a bit of a gray area in the semantics of our discussion however, since they are both actor and critic. An argument could be made therefore, that, despite being trained by multiple teacher value functions, ultimately AMN trains just a single value function (and consequently also a single policy). We will include further discussion in our revised paper however - in addition to more citations to relevant DQN literature, with a current plan to include a more thorough discussion on related work in the paper’s appendix so as to give relevant past literature their appropriate dues. This will also include a brief discussion of the two additional references you mentioned (thanks for those).
>
> Note on why we distinguish strongly between DQN and actor-critic frameworks: As we mention in our initial response to Reviewer kVWj, we believe that, practically, there is a meaningful distinction between actor-critic methods, where policies and value estimations are separate, from DQNs, where both the action policy and the value estimation are encapsulated by a single network. Critically, DQN methods have the practical disadvantage of being limited to discrete action spaces and also needing the full representational and computational complexity of the value estimation to be preserved at inference time after training. As shown by Andrychowicz et al. (“What matters for on-policy deep actor-critic methods? A large-scale study”, ICLR 2021) abd Mysore et al. (“Honey, I shrunk the actor: A case study on preserving performance with smaller actors in actor-critic RL”, COG 2021), oftentimes, it appears that the value function often bears the higher burden of required complexity. Noting this, there is a significant practical advantage to being able to just preserve the smaller, less computationally expensive actor networks of actor-critic methods and disregard the value estimation networks at inference time.
>
> Regarding including results for more actor-focused multi-task improvements, we agree that they would be interesting additions for a discussion about multi-style learning on the whole, and we will consider including those in our revision if we are able to access codes for prior work that we can quickly integrate with our learning and testing framework - though we will prioritize improving the statistical significance of our results and improving the representation of existing results, as suggested by Reviewers eF9d and mdph.
>
> As stated before, please let us know if there are aspects of what we have outlined that are unclear or could warrant further elaboration either here in the paper responses or in the revised paper. We also invite you to offer any additional feedback you may have on the revision we will upload in the coming days. Thank you again for your support of our work.

---

> > ### Comment · Reviewer_Lh1z · 2021-11-21
> > **Interesting details**
> >
> > Thank you for the detailed explanations. Room is always difficult to find in a paper, but I think that what you explain to me deserves to be (in a shorter form) in the paper. I think that the discussion regarding the complexity of critic networks is very interesting. It emphasizes the multi-criteria aspect of RL itself: sometimes, sample-efficiency is not the unique metric to be optimized, but compute efficiency may be. If you could find an industrial setting in which RL is used, and compute efficiency is paramount, it would add much strength to your argument that sometimes, critic-only methods don't need to be compared against.

---

### Official Review · Reviewer_eF9d · 2021-10-25

**Correctness:** 3
**Technical Novelty And Significance:** 3
**Empirical Novelty And Significance:** 2
**Recommendation:** 6
**Confidence:** 4

**Details Of Ethics Concerns:**

No concerns.

**Main Review:**

# Strengths

1) Simple yet novel approach
	- Exploring multiple critics for multiple reward functions for a single task is a natural idea that has not been done before.
	- There are clear ways to take these ideas further when needing to combine behaviors and author/control the behavior of RL agents trained with this method.
2) Intriguing qualitative results
	- The UFC game videos are promising for the way behaviors are distinguished, even if the quantitative analysis is less clear. Without baselines it's hard to know if this would not obtain with other methods, but it's clearly promising.


# Weaknesses

1) Lack of statistical rigor in results
	- The results do not compare algorithms using tests for statistical significance of differences. This makes it hard to tell which results hold up under repeated trials.
	- From a glance the error bars in the Sonic results all overlap for most algorithms (Figure 4).
	- Similar overlaps seem to obtain for Pong (Table 1). For example, the SAC Defensive setting play time of MT w/One-hot is 1446±442, which overlaps the MN-MultiCriticAL 1637±113.
	- The UFC results do not report any measure of variance.
	- Action: Provide statistical testing of these differences. Or consider reporting other metrics recommended for RL evaluation (for example: https://arxiv.org/abs/2108.13264)
2) Lack of baselines in the UFC domain
	- The authors note that the domain is meant for demonstration only and not as a benchmark. I found the qualitative results shown most compelling for this case and so was disappointed to not have baseline comparisons (acknowledging the costs for doing these experiments).
	- Action: Adding these baseline comparisons would be a great improvement to the paper. I recognize this may not be feasible, but want to indicate to the authors that this would be a strong demonstration of the work. An alternative may be to consider some other simple continuous domain (walker, half-cheetah, &c.) that more clearly demonstrates the potential to learn stylized behavior.
3) No direct test of the interference hypothesis
	- One of the more interesting ideas in the paper is that MTRL may suffer from interference between task heads when sharing a backbone. In the methods this is the difference between the MH and MN variants of MultiCriticAL.
	- No tests in the paper or analysis investigate this claim.
	- Action: It would strengthen the paper to show statistical tests that clearly demonstrate this claim.
4) Some missing technical details
	- Below are some minor details that should be easy to address to improve the paper
	1) Why do the single task networks perform so well? Are they only trained and tested in a single task setting? This was not explained in the methodology.
	2) What is the evidence that UFC agent "smoothly transition between the different fighting styles"? The results show different styles but do not directly assess transitioning between styles.
	3) How does one interpret Table 3 given agents were trained on joint parameter combinations, rather than the single dimensions shown? Are these roughly the extreme quadrants and center of a 2D grid?

**Summary Of The Paper:**

The paper introduces a variant of actor-critic reinforcement learning algorithms where multiple critics are trained in a reduced version of multi-task training where a task allows multiple reward functions. Two variants are proposed for handling multiple reward functions: one where a backbone value function is trained with multiple heads and one where completely distinct value functions are trained. Evaluation considers three domains: tracing different shapes, Pong with aggressive-defensive styles, and Sonic the Hedgehog game levels (where levels are styles). Results are ambiguous as to the improvement of the architecture over classical methods. Some qualitative results are shown for a fighting game trained with the multi-critic approach.

**Summary Of The Review:**

The paper presents a technique with promise for training agents to perform tasks in a variety of ways. The current empirical results are ambiguous when comparing the proposed method to baseline methods, leading me to recommend rejection.

My score would increase if the authors can provide clear (statistically) conclusive results of superior performance of the proposed method.

**Update**
I have increased my score in light of the additional experiment seeds added and clearer differences in performance from the resulting reductions in variance.

---

> ### Author Response · Authors · 2021-11-12
> **Initial response to Reviewer eF9d**
>
> Thank you for taking the time to review our paper and for the clear statement of actionable elements to help us improve the quality of our paper.  With this initial response, we present our plan to revise the paper within the next few days to address your concerns and those raised by the other reviewers and also try to address a few of the queries you raised about our work. Please let us know if there are elements you would offer further notes on.
>
> We will provide our responses following the sequence of Weaknesses you addressed in your review (by adding a ‘W’ prefix).
>
> W1. We tried to follow the typical reporting procedures used in recent RL literature, but agree that the presentation can be improved to offer better statistical significance. We will update our results with more seeds and also follow some of the suggested metrics.
>
> W2. Given that the UFC domain uses proprietary code and cannot be easily reproduced, we did not intend to use this as a true benchmark but rather as a proof of utility. We also encountered a few issues related to memory usage in running the environment, making it unsuitable for batch experimentation and therefore additional baselines were omitted in the interest of time for the original submission. For the revision, we have been collecting additional results as well as data for MTPPO to provide a (hopefully) more meaningful comparison and demonstration of utility.
>
> W3. Regarding our inference about the possible interference of multiple tasks/styles sharing a value estimation backbone, we were drawing from the fact that MN-MultiCriticAL (where each critic network is distinct) consistently outperformed MH-MultiCritical (where the multi-headed critics share some representation) as well as the fact that both forms of MultiCriticAL outperformed MTSAC and MTPPO on the multi-style tasks. This, to us, implied that, at least for the tested benchmarks, shared representation in the value estimation may have in fact been acting against policy learning rather than in service of. We hope that improved statistical significance in our updated results will prove more convincing.
>
> W4.2 Yes, the single-task networks are trained and tested on a single task, with no knowledge of other tasks. We noted that they do not always perform best however, as seen with the Sonic games, where the multi-level policies did sometimes outperform the level-specific policies.
>
> W4.3 and 4.4 The styles listed in Table 3 (and Figure 5) of our original submission actually represent combinations, where the ‘Aggressive’ style represents a weight of 1.0 for the aggressive/defensive setting and 0.5 (corresponding to neutral) for the moving/blocking setting. Similarly Defensive is (0, 0.5), Neutral is (0.5, 0.5), Blocking is (0.5, 1.0), and Moving is (0.5, 0.0). We omitted this specificity in the table in an attempt to make it easier to parse, but it seems we may have contributed to increased confusion instead. This will be adjusted in the revision.
>
> We hope that the additional results and analysis we will provide in the revision, which we shall upload within the next few days (and bring to your attention via this openreview discussion) will be convincing enough and that you might reconsider your evaluation. As stated before, please let us know if there are aspects of what we have outlined that are unclear or could warrant further elaboration either here in the paper responses or in the revised paper.

---

### Official Review · Reviewer_mdph · 2021-11-01

**Correctness:** 3
**Technical Novelty And Significance:** 3
**Empirical Novelty And Significance:** 3
**Recommendation:** 6
**Confidence:** 4

**Main Review:**

The proposed ideas are presented clearly and the exposition on related work throughout the paper is useful in guiding the reader.

The main technical contribution is the idea of training separate critics or value heads while maintaining a single, shared actor for multiple tasks. As the authors pointed out in the rebuttal, the choice of a single actor is significant for many compute-sensitive applications (e.g. video games), where inference can be done in parallel (e.g. computing the actions of several NPCs executing different “behaviours” in a single forward pass). I think the paper’s claims are interesting and somewhat surprising since they suggest that negative inference disproportionately affects value learning over policy learning. Furthermore, this challenges the need for the explicit separation of policies for different tasks, which is a common technique in the multi-task RL literature, e.g. [1,2].

The experiments consider several unique, well-motivated environments and tasks. Following the paper’s revisions, I find that the results convincingly demonstrate the claims and address the majority of my initial concerns. However, a comparison against a state-of-the-art multi-actor, multi-critic algorithm could have strengthened the experiments. Currently, the “single-style” baseline is meant to fill this role, however it is difficult to compare this to the authors’ proposed approach in terms of sample efficiency since: (1) the single-style is only trained on a single task vs multiple simultaneous tasks (2) single-style does not observe sample efficiency gains from shared multi-task training.

Additional comments:
- Tables and graphs need to report the meaning of the error margins. i.e. are they reporting standard error, 90% confidence intervals, or something else?
- Typos: “mutli” (second paragraph, first sentence), “mulit” (appendix B header)


[1] Yang, Zhaoyang et al. “Multi-Task Deep Reinforcement Learning for Continuous Action Control.” IJCAI (2017).

[2] Teh, Yee Whye et al. “Distral: Robust multitask reinforcement learning.” NIPS (2017).


**Summary Of The Paper:**

This paper proposes a single-actor, multi-critic approach to address the well-known problem of negative interference in multi-task RL (MTRL). The work focuses on a special case of MTRL described as “multi-style RL”, where the goal is to learn several distinct behaviours under the same environment dynamics. Experiments were conducted on a wide variety of environments: a path following domain, Pong, Sonic the Hedgehog, and a UFC fighting game. Results provide evidence that the proposed approach achieves a better final performance than a single-actor, single-critic MTRL baseline.

**Summary Of The Review:**

Following the rebuttal and the paper revisions, I vote to accept. The paper demonstrates that several distinct behaviours can be successfully learned with a novel single-actor, multi-critic setup. While the technique is simple and the main idea shares some motivation with previous work, I believe this will be a valuable contribution to the multi-task RL literature.

---

> ### Author Response · Authors · 2021-11-12
> **Initial response to Reviewer mdph (part 1/2)**
>
> Thank you for considering our work and providing your review of our paper submission. With this initial response, we hope to address some of your queries and outline how we plan to revise the paper within the next few days to address your concerns and those raised by the other reviewers. Please let us know if there are elements you would offer further notes on.
>
> Firstly, to address the question of motivation. While sample efficiency is an element of why we sought to employ single-actor architectures, it was not the main consideration. Here, we feel it might be important to distinguish the motivation for the paper and the motivation for our work, more broadly speaking. The paper was motivated simply by the fact that we observed a surprising hole in current literature and sought to fill it, seeing as there appears to be an empirical advantage to single-actor, multi-critic actor-critic frameworks that appears to not have been considered in prior work. We did observe benefits to sample efficiency during training, but we framed it primarily as a function of our evaluation metrics after an equivalent number of training samples - if after the same amount of training data, MultiCriticAL policies were able to consistently attain higher performances, it follows that the training is more sample efficient. We do however agree with your identification of the lack of training curves as a weakness of the results presentation in the original paper. We will provide these in the revision, which will include more training seeds to improve statistical significance and some of the more robust evaluation metrics suggested by Reviewer eF9d. This also applies to your second point regarding the representation of our results.
>
> With regards to the broader motivation of the work in general, it was driven by an interest in developing more robust action controllers for characters in video-games, where resource constraints are a very serious consideration. For one, storing multiple actor networks would incur higher memory and storage costs compared to just a single actor. Using multiple actors could also be more computationally inefficient when deployed at scale when controlling multiple characters. Consider, for example, if each game character required a separate learned actor - this would require us to maintain separate actors in memory or move actor networks in and out of working memory. Being able to use a single actor would alleviate some of these concerns and also allow us to take advantage of hardware acceleration of parallel inference in computing the actions for multiple characters at once with a single network. Ultimately, the main motivation here was runtime utility as, in theory, once trained, a lot of the network’s lifetime compute will be dominated by runtime inference, where training sample efficiency is less of a problem than runtime efficiency.
>
> Additionally, when considering acting with a combination of styles, training and maintaining multiple actors can be computationally expensive. For example, Peng et al. (“Deepmimic:  Example-guided  deep  reinforcement  learning  of  physics-based  character  skills”, SIGGRAPH 2018), after running into the limitations of a single-actor, single-critic approach, used a multi-policy approach where actions are taken as a weighted sum over the output from each style policy. Considered more generally, there are no guarantees that actions proposed by each policy would be similar enough that a weighted sum over them would always make sense (destructive interference is possible), as cooperation between policies is not something that is explicitly trained for. One might gain better results through such explicit conditioning, though we are not aware of work exploring such an idea and would imagine cooperation metrics would be complicated to define, depending on the task. Peng et al. seemed to avoid possible action composition issues however, likely due to their training being based on imitation learning, where there are certain similarities of movement already built into the fact that the reference motions follow similar movement dynamics. A simple additional experiment we could include on this would be on the Pong benchmark, where we could compare interpolating between styles with MultiCriticAL (as in Table 2 of our paper) to interpolating between single-style action policies.

---

> > ### Author Response · Authors · 2021-11-12
> > **Initial response to Reviewer mdph (part 2/2)**
> >
> > Regarding the note about comparing to multi-actor, multi-critic approaches, the most naive form of this is what we sought to capture by comparing against single-style policies - where each style has its own policy and critic. We recognize that this does disregard gains that may be gained through shared representations in both the actors and critics, but felt that including multi-actor policies in the discussion may confuse or distract from a discussion on the simple core utility of the multi-critic approach - elements of which may extend to the multi-actor architectures but was outside the scope of our study.
> >
> > Regarding your notes about the relative capacities of the critic being the distinguishing factor, you make a good point and it is one that we considered as well. As we discussed in Section 5.5 of our original submission, we noted that MN-MultiCriticAL seemed to often outperform MH-MultiCriticAL, but we also noted that this came at the increased cost of computing and training multiple separate value networks. The MH-MulitCriticAL architecture however alleviates some of this by only introducing additional heads at the final layer of the network. This is indeed more computationally expensive than computing a single Q-value but not significantly more so, given the relative benefits it still offers. We also noted that increasing the capacity of a single-critic value network may also allow for improved training, but also highlighted the lack of principled techniques to determine a suitable increase in complexity.
> >
> > On the note of hyperparameters, we try to ensure fairness by fixing all common hyperparameters between MultiCriticAL as well as SAC, PPO, MTSAC and MTPPO. These were found by identifying the parameters that consistently enabled training in the single-style settings. You are correct that more tuning may improve results, but what we show is that MultiCriticAL can more readily improve multi-task/multi-style performance.
> >
> > Regarding the performance of single-style actors, we believe that the cases where multi-style actors may perform better (which was only observed for the Sonic games) may have to do with shared representations learned by the policy actually improving the general policy performance. This hypothesis is difficult to test without being able to actually explain the learned policies however and is therefore largely limited to conjecture. For both the Path following and Pong experiments though, we observed that the single-style policies tended to learn better policies than their multi-task counterparts. We can more directly address these observations in the revised submission.
> >
> > We hope that the additional results and analysis we will provide in the revision, which we shall upload within the next few days (and bring to your attention via this openreview discussion) will be convincing enough and that you might reconsider your evaluation. As stated before, please let us know if there are aspects of what we have outlined that are unclear or could warrant further elaboration either here in the paper responses or in the actual paper.

---

> > > ### Comment · Reviewer_mdph · 2021-11-15
> > > **Response to Authors 1**
> > >
> > > Thank you for your comments and clarifications. I'd like to address three points in your latest reply.
> > >
> > > - **re: the motivation for a single actor over multiple actors (one for each task/style)**. You make a good point about memory costs and controlling multiple NPCs using the same, shared policy which I initially overlooked. In particular, with a shared policy, you can compute the actions for multiple NPCs (possibly executing different tasks/styles) in parallel with a single forward pass, and I don't believe this can be easily done with heterogeneous networks for each skill. I think the paper can elaborate on this more to make your point clearer, and preferably in Section 1 instead of Section 3. Currently, Section 1 identifies the "hole" in the literature of single-actor, multi-critic MTRL but without much motivation for the single actor choice.
> > >
> > > - **re: sample efficiency**. I may be misinterpreting the results, but I don't see how the different baselines were trained for the same number of samples, as you claimed. For example, in Pong, single-style PPO was "trained for 150 epochs with ..." and MTPPO was "trained for 250 epochs with ..." (where "..." is the same for both). Could you please clarify? As you've noted, full training curves would greatly improve the readability of results.
> > >
> > > - **re: critic sizes**. In particular, I'm worried that the single-style PPO baseline's critic is far too small (and perhaps that's why it's performing poorly). If I'm understanding correctly, if there are $k$ tasks, the MTPPO critic would have roughly $k$ times more parameters. While I agree it is difficult to compare expressivity across different neural architectures, I think roughly equalizing the number of neurons between baselines (at least for the critics) -- i.e. making single-style PPO's critic $k$ times larger -- would significantly help to show the proposed claims. This would strengthen the hypothesis that multiple critics help mitigate the issue of task interference over a single critic of similar size.

---

> > > > ### Author Response · Authors · 2021-11-17
> > > > **Response to Reviewer mdph response 1**
> > > >
> > > > Thank you for reviewing our initial response and for your followup.
> > > >
> > > > Below are a few quick notes regarding your comments:
> > > >
> > > > **re: being clearer with motivation** Noted. You make a good point about the lack of clarity in how we originally framed our arguments and we will try to make it clearer in the revision.
> > > >
> > > > **re: sample efficiency** The single-style policies, since they only have to learn their one specific style/level/task, typically trained faster than the multi-style policies. We mainly used the single-style policies as a way to establish a performance target for the multi-style policies and not strictly as a baseline - our main baselines were Multi-Task (MT) PPO and SAC, which use a one-hot task/style encoding and a single-headed critic network, as is typical in literature. Our introduced frameworks, Multi-Network and Multi-Headed (MN- and MH-) MultiCriticAL PPO and MultiCriticAL SAC, are trained for the same number of environment interactions as the MTPPO and MTSAC baselines. Since the multi-style policies were required to learn more than just a single style, we allowed them more time to learn the different styles, based on how long they typically took to plateau on the multi-style configurations. We tried to ensure relative training fairness mainly between the MT baselines and our MultiCriticAL extension.
> > > >
> > > > ***Edit***: While revising the paper, we noticed a typo in Appendix D where we said "Multi-style MultiCriticAL-PPO – Trained for 150 epochs ..." - this should have said 250 epochs and was a copy-paste error. If this is what contributed to the confusion, we apologize and are fixing it.
> > > >
> > > > **re: critic sizes** The single-style PPO critic and the MTPPO (or similarly SAC) critics are very similar in size, with the MTPPO critic being slightly larger to accommodate the one-hot style encoding that is added to the state representation. Is it possible you meant to call out the size difference between MTPPO and MultiCriticAL (our framework) PPO? If so, then your note on relative overall critic size would be true for the Multi-Network (MN) MultiCriticAL case, where a separate critic network is learned for each style. In this case, $k$ styles would contribute to a $k$-fold increase in the overall value-representation capacity. In the Multi-Headed (MH) MultiCriticAL case however, we do not have multiple networks, but just multiple heads on the output layer. This does contribute to an increase in the network capacities, but they are relatively small and still contribute to significant performance gains. To speak quantitatively:
> > > > 1. In the *Path Following* task, the MH-MultiCriticAL value representation (including all critic heads) is just about **3% larger** in terms of trainable parameters and, at least in the SAC case, is the difference between learning all 3 shapes and not (Figure 2).
> > > > 2. In the *Pong* task, the MH-MultiCriticAL value functions are **0.3% larger** for PPO and **0.8% larger** for SAC and in both cases contribute to the difference between being able to learn multiple styles and not (Table 1).
> > > > 3. In the *Sonic* games, where we introduce the largest number of heads (17 to correspond with 17 levels), the MH-MultiCriticAL PPO value function is **2.4% larger** and contributes to 20% and 40% gains over MTPPO in Sonic 1 and Sonic 2 respectively (Figure 4 and Section 5.3).
> > > >
> > > > ***Edit 2***: Numbers updated to reflect updated results
> > > >
> > > > It is certainly possible that increasing the MTPPO critic size $k$-fold may improve the performance, but our results show that that is not necessary, as the principled increase in the number of critic heads (each effectively acting as a 'separate' critic) is already sufficient for significant improvement. We are adding details to the appendix of the revision to show the number of and relative differences in trainable parameters for each training framework on each of our benchmark tasks and will also try to provide a clearer discussion on performance gains relative to increased network capacity.

---

> > > > > ### Comment · Reviewer_mdph · 2021-11-21
> > > > > **Thanks for the clarification**
> > > > >
> > > > > Regarding critic sizes, you were correct that I meant to refer to MTPPO (and not single-style) vs MultiCriticAL PPO -- sorry for the confusion. I'm satisfied with all the answers you've provided and I appreciate your responses to my points. I will update the score accordingly (but I would like to see the updated experimental results before doing so).

---

> > > > > > ### Author Response · Authors · 2021-11-23
> > > > > > **Revision submitted - we hope it meets your expectations**
> > > > > >
> > > > > > Please note that we have submitted the revision with the updated results.
> > > > > >
> > > > > > The results on the benchmarks now reflect 15 seeds per task and per algorithm to hopefully provide better statistical significance. We also now report results within the interquartile range to mitigate the impact of outliers (which also reduced the perceived variance in our results) and provide box plots and learning curves in addition to the previously tabulated results.
> > > > > >
> > > > > > As mentioned in our initial response, we also tested composing different styled actors on the Pong task and found that, as we had expected, interpolating between proposed actions from policies with different agendas does result in complete failure to perform, with the composite policy failing to behave at all reasonably in the intermediate style space. While not a fully rigorous evaluation, as it is limited to only one task, we believe it is still illustrative and a brief discussion on this subject is provided in Appendix A.3 of the revised paper.
> > > > > >
> > > > > > We also updated the intro. per your recommendation. We could not make enough room to address all the points we were able to make in our correspondence with you but tried to ensure we captured the most salient points.
> > > > > >
> > > > > > A table of the number of learnable parameters for each critic is also provided in the Appendix.
> > > > > >
> > > > > > For a full list of changes, please refer to [our revision note](https://openreview.net/forum?id=rJvY_5OzoI&noteId=1Ri5kkZxNSk)
> > > > > >
> > > > > > Thank you again for taking the time to review our work and for engaging in discussion with us to help us improve our paper.

---

> ### Author Response · Authors · 2021-11-28
> **Response to updated review post-revision**
>
> Thank you for taking the time to reevaluate our work. Improving the paper was helped a lot from receiving clear and actionable feedback.
>
> Apologies for not responding sooner to your updated review notes - it appears openreview does not email notifications about review edits.
>
> We agree that the evaluations may have been stronger with additional baselines that consider more actor-side improvements. Overall, we hoped for this work to initially establish the base efficacy of single-actor, multi-critic frameworks, since they have been studied less, and we wanted to start filling the previously mentioned 'hole' in literature. We tried to preserve a notion of fairness with like-for-like comparisons and therefore focus initial comparison mainly on MTPPO and MTSAC. That said, a more exhaustive study on which actor improvements may better enable multi-style learning, where per-style strategies may conflict, and how such improvements would interact with multi-critic frameworks, would be interesting and we believe it would make for valuable future exploration.
>
> ***Re: additional comments***:
> - We are reporting a 95% confidence interval and this will be made clearer in the next version of the paper
> - Thank you for catching the typos

---

### Official Review · Reviewer_kVWj · 2021-11-02

**Correctness:** 3
**Technical Novelty And Significance:** 2
**Empirical Novelty And Significance:** 3
**Recommendation:** 8
**Confidence:** 4

**Main Review:**

Strength:
1) The problem of multi-objective RL has many important application in the real world, such as games and robotics. This paper tackles the exact challenges in multi-objective RL.
2) The proposed algorithm is simple, but effective.
3) The examples in the evaluation (e.g. Pong and the boxing game) demonstrates an useful application of the proposed method: designing AI for games.

Weakness:
1) One important reference in multi-objective RL is missing: [1] "Multi-Objective Reinforcement Learning using Sets of Pareto Dominating Policies, Van Moffaert et al., JMLR, 2014". The proposed algorithm is very similar to Section 2.2.1 in that paper. The difference is that this paper extends the same multi-critic-single-actor idea to Deep Reinforcement Learning. However, the claim in this paper that "seemingly no prior work explores the use of a single actor with multiple critics" is not true. The existence of [1] significantly reduced the novelty and technical contributions of this paper.

Additional questions and comments:
1) To train the multi-critic-single-actor (eq. 5, 6, 7, 8), do you need to specify a fixed set of weights w? Or are the w randomly sampled during the training stage?

2) Most of the results are shown in a tabular form. It would be great to visualize the learning process with learning curves. Learning curves gives a lot more insights than the final reward, such as learning speed, sample efficiency, and the progress of learning different styles. It could also be a way to validate the claim that the proposed algorithm mitigates the negative interference between tasks.

3) Section 5.3, "MTPPO agents perform worse on average than agents trained on single-levels". Are the agents "trained on single-levels" just trained on one out of the 17 levels, or trained on all levels but the policy does not take the one-hot encoding of the level as an additional input?

**Summary Of The Paper:**

This paper proposes to extend the actor-critic frame to tackle multi-objective (multi-task) reinforcement learning. The key idea is to learn multiple critics that correspond to different reward functions. Then a single policy is optimized for a weighted combination of these critics. The paper applies the multi-objective RL to learning different styles of completing tasks, such as aggressive or defensive style in a boxing game. The paper shows that the proposed algorithm can beat several multi-task learning baselines.

**Summary Of The Review:**

The paper proposes a simple and yet effective algorithm for multi-objective reinforcement learning. The paper is well written and the results are convincing. However, it misses an important prior work that has similar high-level ideas. The examples in the Evaluation Section of this paper show its great potential in real-world applications, especially for designing AI in video games.

---

> ### Author Response · Authors · 2021-11-12
> **Initial response to Reviewer kVWj (part 1/2)**
>
> Thank you for taking the time to review our paper and for your favorable leaning towards our work. Thank you as well for highlighting elements of our work that require further clarification. With this initial response, we would like to address your additional questions and make a note regarding a weakness of this paper that you noted in your review. This response will outline some of the actions we will take to improve our paper during the rebuttal period so please let us know if there are elements you would offer further notes on.
>
> First, we will try to address your questions and comments:
>
> 1. The weights can be fixed or sampled (either randomly, or following some other sampling protocol) as per the task design. In our experiments, the weights correspond to one-hot encodings for path following and the sonic games - where we did not think it necessarily made sense to have an intermediate space between tasks that we would care about. For Pong, we consider both the one-hot approach and randomly sampled weighting cases - see Table 2 of our paper where binary style selection corresponds to using the weights as one-hot indicators, and explicit interpolation involved randomly sampling an aggression weight, $w_a$, in the range of [0,1] where the defensive weight, $w_d$ was set as $1 - w_a$. The UFC case involved randomly sampling style weights for each episode, similar to the explicit interpolation case for Pong. In hindsight, this was not made clear in our paper and we will endeavour to clarify it in the revision.
>
> 2. Thank you for the note.  Reviewer mdph also noted that as an omission to address. We can provide additional figures to provide a visualization of learning efficiency and efficacy in our revision. We are also collecting additional data to improve the statistical significance of our results.
>
> 3. The single-level agents are trained and tested on just 1 out of the 17 levels in the game. They do not use the one-hot encoding as it would not be necessary in that case, since they are never exposed to the other levels. If you felt this was unclear, we can try to make this clearer in our revised submission.

---

> > ### Author Response · Authors · 2021-11-12
> > **Initial response to Reviewer kVWj (part 2/2)**
> >
> > Regarding your note about the JMLR paper by Van Moffaert et al., we agree that this is a relevant part of prior literature and needs to be cited in our work (and will be in the revision). We would offer though that the key difference in this work is not in that we applied it to the deep RL case, but rather specifically to deep actor-critic RL.
> >
> > As noted by Reviewer Lh1z, multiple Q functions have been used in single-task RL for improving stability of learned value functions in DQN architectures such as with Bootstrapped DQNs (“Deep exploration via bootstrapped DQN”, Osband et al., 2016), Bootstrapped Dual Policy Iteration (“Sample-Efficient Model-Free Reinforcement Learning with Off-Policy Critics”, Steckelmacher et al., 2019), Dueling Q networks (“Dueling Network Architectures for Deep Reinforcement Learning”, Wang et al., 2015), Double Q-learning (Hasselt et al. 2010; “Deep reinforcement learning with double q-learning”, Hasselt et al., 2016). Fedus et al. also do an interesting thing where they consider multiple Q-value approximations for hyperbolic discounting (“Hyperbolic discounting and learning over multiple horizons”, 2019) of value estimations over different time horizons.
> >
> > There are also applications in multi-task settings that, in effect, learn task-specific Q values for a single policy in DQN architectures (“Progressive Neural Networks”, Rusu et al, 2016; “Policy Distillation, Rusu et al. 2016; “Sharing knowledge in multi-task deep reinforcement learning”, D’Eramo et al., 2020). In the case of progressive neural nets and Multi DQN (D’Eramo et al.), there are separate sets of Q-heads for each task, but since DQNs take actions based on the argmax over the Q values, one could also make the argument that they are multi-actor networks. In the case of policy distillation, or similarly Actor-mimic (Parisotto et al.), ultimately, multiple actors need to be trained first before supervised training is used to distill information into a single actor, so these might be gray areas in terms of classification.
> >
> > What we meant to claim, when saying that it seems there is no prior literature exploring single-actor, multi-critic learning, we were referring specifically to actor-critic methods, which learn policy and value networks separately. We believe that, practically, this is a meaningful distinction from DQNs - where both the action policy and the value estimation are encapsulated by a single network (or perhaps more of just the latter since the action taken is the argmax of the learned values). Critically though, DQN methods have the practical disadvantage of being limited to discrete action spaces and also needing the full representational and computational complexity of the value estimation to be preserved at inference time after training. As shown by Andrychowicz et al. (“What matters for on-policy deep actor-critic methods? A large-scale study”, ICLR 2021) abd Mysore et al. (“Honey, I shrunk the actor: A case study on preserving performance with smaller actors in actor-critic RL”, COG 2021), oftentimes, it appears that the value function often bears the higher burden of required complexity. Noting this, there is a significant practical advantage to being able to just preserve the smaller, less computationally expensive actor networks of actor-critic methods and disregard the value estimation networks at inference time. Given that our work is largely motivated by application to video games, where runtime resource use is an important consideration, this was an important distinction.
> >
> > Elements of this discussion were cut from the submitted paper due to space considerations, but we can reintroduce it in the revision, and perhaps include a more thorough discussion in the appendix. We are not claiming high levels of theoretical novelty with our work but rather simply mean to highlight a hole in existing literature on how multi-task RL may be practically realized.
> >
> > We will upload a revision within the next few days (and bring to your attention via this openreview discussion), which we hope you will consider in reevaluating our paper after this rebuttal period, and as stated before, please let us know if there are aspects of what we have outlined that are unclear or could warrant further elaboration either here in the paper responses or in the actual paper.

---

> > > ### Author Response · Authors · 2021-11-23
> > > **Note to Reviewer kVWj regarding submitted revision**
> > >
> > > Please note that we have submitted a revision of our work.
> > >
> > > In response to yours and the other reviewers' reviews, we have added additional discussion around how multiple Q values have been used in other contexts of RL as well as in multi-task DQN architectures. We also make note of what we perceive to the operative differences between prior work and ours. This now also includes a reference and brief discussion around the work by Van Moffaert et al., which you brought to our attention.
> > >
> > > We hope you were satisfied with our responses to your questions and comments and that you might consider raising your recommendation regarding our paper.
> > >
> > > Thank you again for taking the time to review our work.

---

> > > > ### Comment · Reviewer_kVWj · 2021-11-30
> > > > **Thanks for your response**
> > > >
> > > > I have read the response and the revised paper. The revision addressed my major concern of this paper: lack of discussions about multi-Q based RL techniques. For this reason, I have bumped up my review score to "8: accept, good paper".

---

> > > > > ### Author Response · Authors · 2021-11-30
> > > > > **Thank you for considering our revision**
> > > > >
> > > > > Thank you for taking the time to reevaluate our work and we sincerely appreciate your vote in support of the paper's acceptance.

---

### Official Review · Reviewer_Y35C · 2021-11-02

**Correctness:** 3
**Technical Novelty And Significance:** 2
**Empirical Novelty And Significance:** 2
**Recommendation:** 6
**Confidence:** 5

**Main Review:**

strengths:

1. The method is straightforward, extending the existing deep RL method via multiple value networks.

weaknesses:

1. The method can not generate different styles under the same environment, or put another way, this method can not generate different styles under the same reward function.  We only have one task or environment in many practical applications, but we want to generate behaviors with different styles. However, this paper can not generate multi-style behaviors when receiving the same reward signal. The algorithm proposed in this paper is more like a multi-task algorithm instead of a multi-style algorithm.

**update**
I have read the response of the authors and increased my score. I still think the title of this paper is misleading.

**Summary Of The Paper:**

This paper wants to propose a simple method to deal with multi-task (style) problems.

**Summary Of The Review:**

This paper proposed a simple method for multi-task problems. The experimental results are convincing, but the usage of this method is limited. Because this method can only be used for environments with multiple reward functions, and their approach can not generate multi-style behaviors for the same task. I think it would be more reasonable for the author to reformat their paper as a multi-task paper.

---

> ### Author Response · Authors · 2021-11-12
> **Initial response to Reviewer Y35C**
>
> Thank you for taking the time to review our work. In this initial response, we were hoping to get some clarification on your points regarding the weakness of our paper, so that we may take the appropriate steps to address them in our revision and also perhaps provide a more directed rebuttal.
>
> To your point about framing this work as addressing a multi-task RL (MTRL) problem, we do broadly agree with you, which is why our exploration of prior work and the baselines we consider are drawn from prior MTRL literature.
>
> MTRL comes in many flavors and encompasses a wide array of possible task configurations, including cases where RL agents need to learn to handle multiple distinct tasks with dissimilar observation and action spaces. The problem of ‘multi-style’ RL that we specifically focus on in this paper is a special case of MTRL, where the environments, dynamics and core goals for the ‘tasks’ remain generally identical, but where there are nuances that distinguish each ‘task’ (the style), which we capture with different reward signals. In effect, this can be considered as requiring agents to learn one broader task, but with different styles. This is discussed in our paper in Sections 1, 3 and partly in 4.
>
> As you rightly pointed out, our study assumes that each unique style has an associated reward signal, in addition to the main overall task reward. Consistently generating different styles of behavior with RL, where each style is pre-defined, to our knowledge, requires some form of appropriate learning signal to inform the learned value functions for each style. However, we only require reward shaping for a few unique styles, and have demonstrated the ability to interpolate between the unique styles in different combinations without additional reward shaping for each combination of styles. Thus, our method is able to express a larger number of different end styles without requiring reward shaping for each end style. While this may not constitute learning separate styles from a single reward exactly, it does demonstrate an ability for our method to handle multiple styles with a more limited set of rewards.
>
> Our definition of style extends from a similar notion of behavior style employed by Peng et al. in “Deepmimic:  Example-guided  deep  reinforcement  learning  of  physics-based  character  skills” (SIGGRAPH 2018), whose work focused on different movement styles for animation tasks, where each style was conditioned through a reward signal which considered the similarity between the policy’s behavior and reference motion data. Our work assumes this style-based reward signal comes instead through reward shaping. One could also employ imitation techniques as in Deepmimic, or alternatively techniques such as inverse RL or behavior cloning, though ultimately still requiring some form of signal (whether in the form of rewards, demonstration or supervision) to define distinct styles. The multi-critic architecture could be extended to such cases, though specifics would depend on how learning signals are provided.
>
> If you perhaps disagree with how our work and Deepmimic consider a notion of RL behavior ‘style’ as a special case of a task, or believe it might conflict with some other prior definition, could you provide references to relevant literature or elaborate on what we might do to more clearly define our problem focus? We can then work to revise our submission accordingly and would request your attention towards our paper again to reconsider if it meets the bar for acceptance.

---

### Author Response · Authors · 2021-11-23
**Rebuttal Revision Submitted**

### To Reviewers and Program Committee

Please be informed that we have just uploaded a revision to our paper following feedback from the reviewers.

For convenience of re-evaluation, we have colored changes made in this revision in blue. This can be changed back to black for the final version.

Please see below for a summary of changes made to the paper in this revision

#### **Major changes:**
- To help address concerns raised about the statistical rigor of our experiments, we conducted additional experiments, raising the total number of seeds tested on our benchmarks to *15 seeds* (up from 5). If more are deemed necessary, we can acquire additional data for a final version of this paper. We also now report metric stats from within the interquartile range to better deal with outliers.
- Results which were previously only tabulated are also now accompanied by corresponding box plots to better visualize the range of values. Additionally, the Sonic the Hedgehog results in Figure 4 are now represented by box plots instead of histograms to provide a clearer representation of the rewards achieved by agents during testing.
- While the UFC task is meant to be mostly illustrative and not treated as a full benchmark (due to it requiring access to proprietary code, thus limiting usability), we agree with the reviewers that we would make a stronger case for our work by demonstrating how our method compares to baselines on this task. To this end, we compare MultiCriticAL PPO against MTPPO in Section 5.4 (where we previously only showed how MultiCriticAL performs) and show that MultiCriticAL is indeed better at learning delineated styles and offers more balanced behavior in the intermediate space between style extremes (despite rewards only being defined at the extremes).
- Learning curves for each of the 3 benchmark tasks in this paper are provided in Appendix A. We hope this helps make a stronger case for the learning efficiency of our proposed MultiCriticAL method over the more traditional single-critic MTRL approach.
- The Introduction has been updated to better reflect the concerns of practical utility at runtime that motivated our work. While we were not able to make enough space to address all the details we provided in our responses to Reviewer mdph, we tried to hit all the main points and hope the revised Intro. offers a clearer elaboration of the motivations of this work.
- Reviewers noted similarities of our work to methods proposed in prior DQN literature. We maintain that we believe there is a meaningful practical difference between DQNs and Actor-Critic methods but agree that these connections bear mentioning in the paper. A paragraph has been added to Section 4 to briefly address these, with further discussion provided in Appendix B. This discussion attempts to consolidate points raised in discussion with the reviewers to give potential readers a better context for both our work and how it relates to existing literature.
- Additional details about the the number of learnable parameters for each task's critics are provided in Appendix E. With this, we can show that the Multi-headed (MH-) variant of MultiCriticAL is only marginally larger (< 3%) than the single-valued typical multi-task (MT) critics but significantly outperforms it in all tested tasks.
- We address the utility (or lack thereof) of *one* possible approach to policy composition for interpolating between distinct single-style policies on the Pong benchmark task in Appendix A.3. It illustrates one of the points we raised about how composing multiple pre-trained policies may not always work if the policies interfere with each other and it mostly serves to demonstrate one possible failing. We do not feel however that it is sufficiently rigorous to base stronger claims on and thus leave it just as a section in the appendix for any interested readers.

#### **Minor Changes**:
- Assorted typo fixes and slight word changes to fit in page limit.
- Rearranged last table and figure to fit within limit.

As the rebuttal deadline is almost upon us, we are unlikely to be able to make any more significant revisions within the next few hours, but will endeavor to respond to any additional comments or feedback given in response to the revision.

***Edit:*** Updated to fix typo in data entry

---

### Decision · Program_Chairs · 2022-01-20

**Decision:**

Accept (Poster)

**Comment:**

In this paper, the authors investigate a multi-task RL actor-critic technique, where a single actor is used while multiple critics are trained (one per task, where each task corresponds to a different reward function). Experiments on several environments demonstrate that this method works quite well in practice.

All reviewers found the proposed approach sensible and effective, in spite of its simplicity. The main concerns were:
- Lack of novelty: although this is indeed not a particularly original idea, the specific instantiation in the actor-critic setup is novel and well motivated
- Some confusing / unconvincing experimental results: after receiving this feedback, the authors were able to upload a new revision that addressed the main concerns
- Focusing on the "multi-style" aspect when this is essentially a multi-task algorithm: although I agree that framing it as a specific case of multi-task learning would make sense and would probably make more appealing to multi-task RL researchers, I do not consider this to be a major issue

In spite of being a relatively straightforward paper, I believe it is good to have strong empirical evaluation of such basic techniques disseminated to the research community, and I thus recommend acceptance, in accordance with reviewers' recommendations after the discussion period.